# Functional alterations of the prefrontal circuit underlying cognitive aging in mice

Huee Ru Chong[1,3], Yadollah Ranjbar-Slamloo[1,3], Malcolm Zheng Hao Ho[1,2], Xuan Ouyang[1] & Tsukasa Kamigaki [1] ✉

Executive function is susceptible to aging. How aging impacts the circuit-level computations underlying executive function remains unclear. Using calcium imaging and optogenetic manipulation during memory-guided behavior, we show that working-memory coding and the relevant recurrent connectivity in the mouse medial prefrontal cortex (mPFC) are altered as early as middle age. Population activity in the young adult mPFC exhibits dissociable yet overlapping patterns between tactile and auditory modalities, enabling crossmodal memory coding concurrent with modality-dependent coding. In middle age, however, crossmodal coding remarkably diminishes while modality-dependent coding persists, and both types of coding decay in advanced age. Resting-state functional connectivity, especially among memory-coding neurons, decreases already in middle age, suggesting deteriorated recurrent circuits for memory maintenance. Optogenetic inactivation reveals that the middle-aged mPFC exhibits heightened vulnerability to perturbations. These findings elucidate functional alterations of the prefrontal circuit that unfold in middle age and deteriorate further as a hallmark of cognitive aging.

Cognitive aging is the natural and gradual decline in cognitive function that occurs as people age and emerges as a major challenge for maintaining quality of life and employment. It is hence crucial to understand the neurobiology underlying cognitive aging in detail. Working memory (WM) decline is a fundamental aspect of cognitive aging and has the earliest onset among age-related cognitive deficits[1–5].

The prefrontal cortex (PFC) is known to play a vital role in executive function[5–12]. Especially in the context of working memory, the PFC and nearby frontal areas typically show sustained activity during memory-guided behavior, implicating their crucial involvement in active maintenance of task-relevant information[13–27]. Moreover, lesion or inactivation of the PFC and its surrounding areas severely impairs memory-guided behavior, supporting the notion that they make a critical contribution to memory maintenance with their sustained activity[15,18,19,21,24–26,28–31].

In line with this, neuroimaging studies comparing young and older adults showed that age-related WM decline is associated with functional deficiency in the PFC and the altered connectivity between the PFC and the task-relevant areas[3,4,32–35]. At the single-cell level, pyramidal neurons in the monkey PFC display age-associated retardation in axon fiber tracts, apical dendritic regression, loss of synapses, and decrease in spine number[36–38], which is predictive of a decline in WM[37]. Additionally, a monkey neurophysiology study found that single PFC neurons exhibit a decline in sustained activity with age during an oculomotor delayed response task, demonstrating that the age effect on WM-related neuronal processing is evident at the level of single-neuron activity[39]. However, it is still unclear how aging alters the functional organization in the PFC for memory maintenance at the circuit level.

To address this issue, we employed calcium imaging and optogenetic manipulation techniques on mice of various ages while they engaged in a memory-guided behavior involving both tactile and auditory stimuli. We showed that with the progression of age, there was a decrease in the fraction of action-plan coding neurons and the

[1]Neuroscience & Mental Health, Lee Kong Chian School of Medicine, Nanyang Technological University, Singapore 308232, Singapore. [2]IGP-Neuroscience, Interdisciplinary Graduate Programme, Nanyang Technological University, Singapore 308232, Singapore. [3]These authors contributed equally: Huee Ru Chong, Yadollah Ranjbar-Slamloo. ✉e-mail: tsukasar@ntu.edu.sg

action-plan signal of individual neurons in the medial prefrontal cortex (mPFC), leading to reduced decodability of the action plan at the population level. In the young adult mPFC, population activity displayed dissociable yet overlapping patterns between the two modalities, enabling concurrent modality-general and modality-dependent coding. Unexpectedly, in middle age, modality-general coding significantly diminished while modality-dependent coding persisted, and both forms of coding were disrupted in advanced age. We also found that resting-state functional connectivity among action-plan-coding neurons decreased as early as middle age. Furthermore, the middle-aged mPFC exhibited higher susceptibility to optogenetic perturbation. Taken together, our findings show that the functional alterations in the prefrontal circuit responsible for memory-guided behavior are already manifested in middle age.

## Results

### Aging retards task learning

To examine the impact of aging on memory-guided behavior and the underlying functional organization in the medial prefrontal cortex (mPFC), we trained head-fixed mice of three different age groups including young (3–8 months; $n = 11$ mice; 6 males, 5 females), middle age (11–14 months; $n = 7$ mice; 3 males, 4 females), and advanced age (18–23 months; $n = 5$ mice; 2 males, 3 females) on a bimodal delayed two-alternative forced-choice task (d2-AFC) using tactile and auditory stimuli (Fig. 1a). Each trial started with the presentation of a sample stimulus (sample period, 500 ms), which included a tactile and auditory compound stimulus that instructed left (2 kHz tone with airpuff to the left cheek) or right (8 kHz tone with airpuff to the right cheek) (referred to as a "Left" or "Right" trial, respectively). This was followed by a 2 s delay period in which the stimulus was absent. The test period then began when the monitor turned white signaling that it was time to respond with a lick; licking the correct-side water port within the 2 s response window was rewarded, and licking the incorrect side was punished with a beep sound and a longer inter-trial interval. While mice in all age groups learned the bimodal d2-AFC task, the middle-aged and the advanced-aged mice required more sessions than the young mice (Fig. 1b and Supplementary Fig. 1a, b) to reach behavioral criteria (70% correct or higher in two consecutive daily sessions), indicating that learning of the task slowed with age.

### Memory coding degrades with aging

After the mice learned the bimodal task, we performed microendoscopic calcium imaging from pyramidal neurons in the mPFC. In the imaging sessions, the young and the middle-aged mice showed comparable behavioral performance (Fig. 1c, $U = 103$, $p = 0.93$, Wilcoxon rank-sum test), while the advanced-aged mice exhibited a slight decrement in performance compared with the middle-aged mice ($U = 59$, $p = 0.030$). They did not show any age-related response time slowing (Supplementary Fig. 1c).

For selective labeling of pyramidal neurons, we injected Cre-dependent adeno-associated virus (AAV) expressing the calcium indicator GCaMP6f[40] into the mPFC of CaMKIIα-Cre mice (Fig. 2a–c and Supplementary Fig. 2a–c). Imaging was performed through a gradient refractive index (GRIN) lens coupled to a miniaturized integrated fluorescence microscope[21,41,42], which allows simultaneous monitoring of activity from multiple single neurons. We found that a subset of pyramidal neurons in the mPFC exhibited action-plan selectivity during the delay period (referred to as 'action-plan-selective delay' neurons) (Fig. 2d), consistent with the previous studies[13–16,18,19,21,23–27]. With aging, however, the fraction of the action-plan-selective delay neurons diminished (Fig. 2e for the early and late delay period; Supplementary Fig. 3a for all the task periods), along with a reduction of the absolute selectivity index of action plan ($|SI|$) during the delay (Fig. 2f for the early and late delay; Supplementary Fig. 3b for all the task periods) in line with the previous monkey electrophysiological study[39]. Consequently, decodability of action plan decreased with age ($p < 0.001$, two-way ANOVA with Bonferroni post hoc test, using 300 randomly selected neurons for each age group, Methods; Fig. 2g), pointing to an age-dependent decline in memory coding in the mPFC.

The age-related decline in the action-plan signal cannot be explained by mere variations in licking behavior during the delay period, as there were no significant differences among the age groups

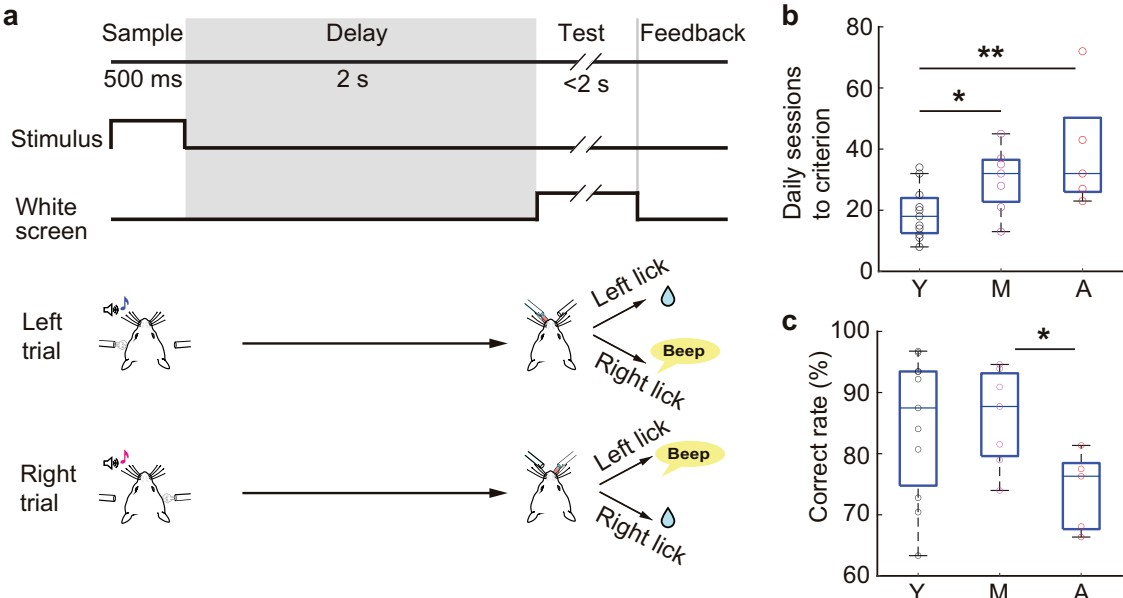

**Fig. 1 | Aging effects on behavioral performance of the bimodal delayed 2-AFC task. a** Schematic of the task design. Mouse cartoons modified from the drawing[73]. **b, c** Number of daily sessions to reach a behavioral criterion (**b**) and correct rate during the imaging sessions (**c**) ($n = 11$ young mice, $n = 7$ middle-aged mice, $n = 5$ advanced-aged mice). Open circles, individual mice. *$U = 81$, $p = 0.032$, **$U = 72.5$, $p = 0.015$, in (**b**), *$U = 59$, $p = 0.030$ in (**c**), Wilcoxon rank-sum test. Box plots show the median and the 25th and 75th percentiles as box edges and the whiskers extend to the 5th and 95th percentiles. Y young; M Middle age, A advanced age. Source data are provided as a Source Data file.

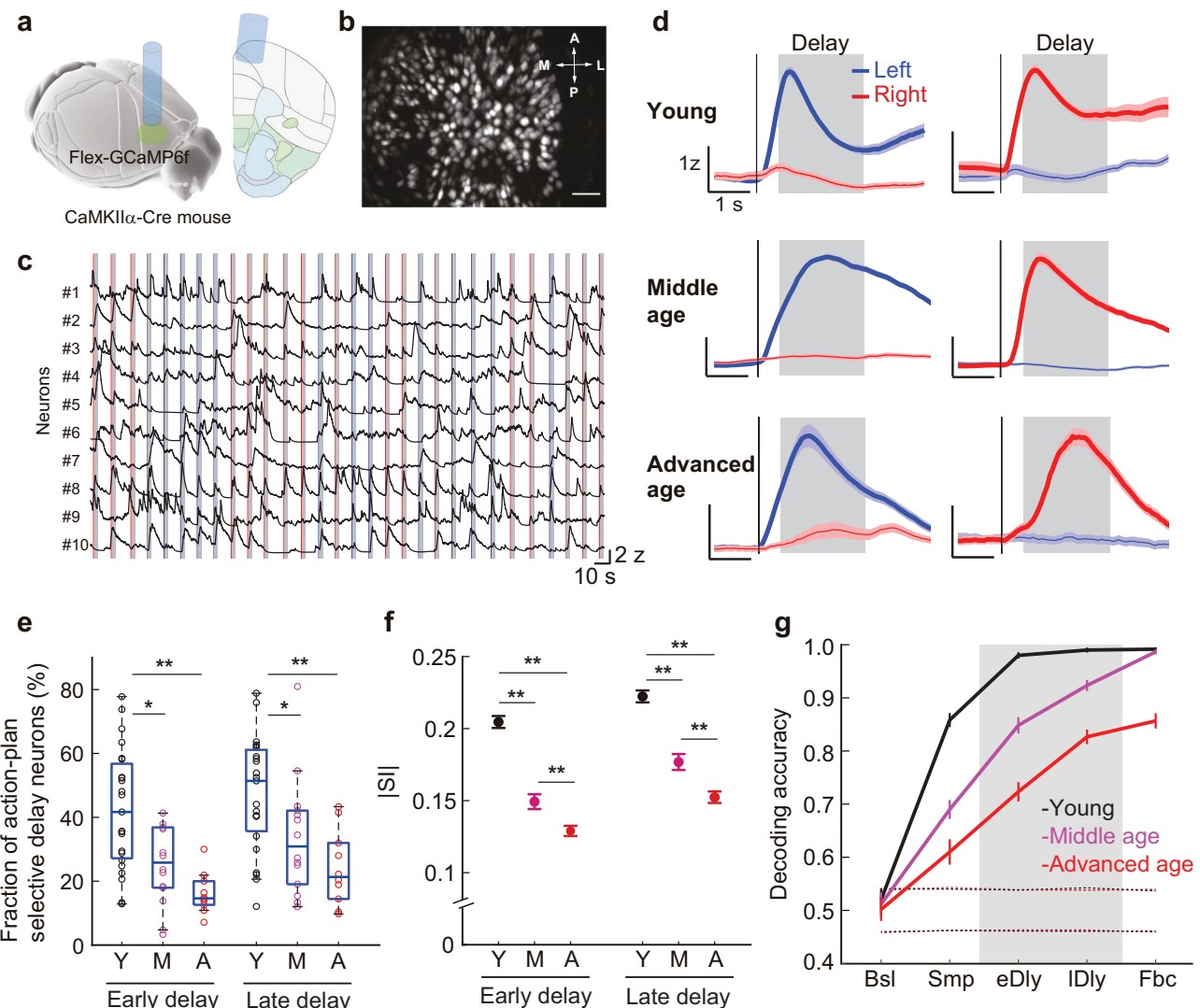

**Fig. 2 | Aging effects on action-plan coding in the mPFC during the bimodal d2-AFC task. a** Calcium imaging with the GRIN lens. Left, modified from the 3D Brain Explorer Allen Reference Atlas[74], atlas.brain-map.org. Right, modified from Allen Mouse Brain Coronal section (image 38)[74], atlas.brain-map.org. **b** Example field of view (maximum intensity projection of dF/F movie). Scale bar, 100 μm. **c** Activity trace of 10 example ROIs extracted from (**b**). Blue and red vertical lines, sample onset in Left and Right trials, respectively. Gray shading, delay period. **d** Trial-averaged activity (z-scored) of the example action-plan-selective delay neurons from each age group. Gray shading, delay period; vertical line, sample onset. Left column, Left-delay neurons; right column, Right-delay neurons. Colored shadings, ± s.e.m. **e** Fraction of action-plan-selective delay neurons for the imaging sessions (young, $n = 23$ sessions, 11 mice; middle age, $n = 14$ sessions, 7 mice; advanced age, $n = 10$ sessions, 5 mice). Left, early delay period, $*U = 504$, $p = 0.037$,

$**U = 485$, $p = 2.5 \times 10^{-4}$; Right, late delay period, $*U = 505$, $p = 0.035$, $**U = 472$, $p = 1.6 \times 10^{-3}$, Wilcoxon rank-sum test. Box plots show the median and the 25th and 75th percentiles as box edges and the whiskers extend to the 5th and 95th percentiles. Y, young; M, middle age; A, advanced age. **f** Mean absolute selectivity index for action plan (|SI|) (young, $n = 2{,}245$ neurons; middle age, $n = 859$ neurons; advanced age, $n = 968$ neurons). Error bars, ± s.e.m. $**p < 0.001$, one-way ANOVA with Bonferroni post hoc test. **g** Decoding accuracy of action plan in each task period. Decoding accuracy during the early and late delay (gray shading) in the middle-aged and the advanced-aged groups was significantly lower than that in the young group ($p < 0.001$, two-way ANOVA with Bonferroni post hoc test). Error bars, ± s.e.m. across 100 iterations. Horizontal dotted lines, 97.5th and 2.5th percentiles of the shuffled data. Bsl baseline, Smp sample, eDly early delay, lDly late delay, Fbc feedback period. Source data are provided as a Source Data file.

regarding this behavior ($F_{(2,20)} = 0.23$, $p = 0.80$ for lick rate and $F_{(2,20)} = 0.08$, $p = 0.92$ for inter-lick interval, one-way ANOVA; Supplementary Fig. 4a, b). To further assess the potential impact of licking on neural activity, we applied a generalized linear model (GLM) to each neuron including lick responses in addition to all other task events (sample, early delay, late delay, feedback) as predictor variables (see Methods for details). Consistent with the original fraction of action-plan-selective delay neurons (Fig. 2e), we found an age-related decrement in the fraction of neurons with significant modulation by the early-delay and late-delay variables, in which lick-related variables were regressed out (Supplementary Fig. 4c). We further tested whether left and right lick rates were already different during the delay period and

found a significant bias to left licking on left trials, and right licking on right trials ($F_{(1,40)} = 34.2$, $p = 7.8 \times 10^{-7}$ for left trials and $F_{(1,40)} = 19.8$, $p = 3.9 \times 10^{-6}$ for right trials; the main effect of licking direction, two-way ANOVA with licking direction × age group; Supplementary Fig. 4d). However, no significant differences were found among the age groups ($F_{(2,40)} = 0.16$, $p = 0.86$ for left trials and $F_{(2,40)} = 0.34$, $p = 0.72$ for right trials; the main effect of age group, two-way ANOVA with licking direction × age group), nor were there any interactions between the age groups and the licking directions ($F_{(2,40)} = 0.11$, $p = 0.90$ for left trials and $F_{(2,40)} = 0.38$, $p = 0.68$ for right trials). This indicates that while premature licking was biased already during the delay period, the bias was not age-dependent. To further investigate whether such

premature licking bias could account for age-dependent delay activity, we specifically analyzed trials without premature licking during the delay period. We observed a decline in decoding accuracy during the delay as a function of age in the bimodal task (Supplementary Fig. 4e), which is consistent with the findings in Fig. 2g. These results indicate that although there was a premature bias during the delay period, it was not responsible for generating the age-dependent delay activity. Overall, these results demonstrate that the action-plan signal during the delay period degraded with aging, and is unlikely to be due to variations in licking behavior among the different age groups.

One could argue that aged mice carry more weight and have heightened water requirements, and thus the relative or subjective reward size is smaller compared with young mice. These variations in the relative value of rewards could potentially exert an influence on both behavior and delay activity. To examine this across different age groups, we conducted experiments employing a tactile-based d2-AFC task with two reward size conditions: default and high reward conditions. In the high reward condition, the reward size was twice that of the default condition. Despite the variations in reward size, we did not observe any systematic changes in behavioral performance across all age groups ($F_{(1,42)} = 0.68$, $p = 0.41$, the main effect of reward size factor, two-way ANOVA with reward size × age group; Supplementary Fig. 5a). Also, there was no significant effect of age group ($F_{(2,42)} = 0.83$, $p = 0.44$) nor interaction ($F_{(2,42)} = 0.64$, $p = 0.53$). We suggest that the default reward size was sufficient to motivate the animals in all age groups to actively engage in the task, rendering any additional increase in reward size ineffective in further improving behavioral performance. Furthermore, we examined whether the variations in reward size had an age-dependent effect on the action-plan selectivity of the neurons. Our analysis revealed that the absolute selectivity index (|SI|) during the delay period did not exhibit any significant dependence on reward size across all age groups ($F_{(1,3769)} = 1.2$, $p = 0.27$ and $F_{(2,3769)} = 0.66$, $p = 0.51$ for early delay, and $F_{(1,3769)} = 0.97$, $p = 0.32$ and $F_{(2,3769)} = 0.10$, $p = 0.90$ for late delay, the main effect of reward size and its interaction with age group, two-way ANOVA; Supplementary Fig. 5b). Nonetheless, there was a notable age-dependent decrease in |SI| ($F_{(2,3769)} = 43.4$, $p < 10^{-5}$ for early delay and $F_{(2,3769)} = 46.0$, $p < 10^{-5}$ for late delay, the main effect of age group; two-way ANOVA), consistent with our original findings. These findings suggest that the age-dependent alteration in the action-plan selectivity is unlikely to be solely attributed to variations in reward processing.

We next examined the possibility that age-related variations in GCaMP6f signal or expression levels could contribute to the observed age-dependent activity. The fluorescent intensity of GCaMP6f in histology sections exhibited no significant difference among the age groups ($F_{(2,20)} = 0.01$, $p = 0.99$, one-way ANOVA; Supplementary Fig. 2d). Likewise, the amplitude of ΔF/F of calcium events during the resting state displayed no age-dependent differences ($F_{(2,40)} = 0.28$, $p = 0.76$, one-way ANOVA; Supplementary Fig. 2e). Moreover, the density of GCaMP6f-positive cells in histology sections showed no significant difference among the age groups ($F_{(2,20)} = 0.07$, $p = 0.93$, one-way ANOVA; Supplementary Fig. 2f). These results collectively indicate that differences in the expression levels of GCaMP6f, fluorescent signal amplitude, or the density of infected neurons are unlikely to account for the age-dependent deteriorations observed in the task-relevant activity.

To further investigate whether the GCaMP6f signal changes observed in calcium imaging represent similar firing rate changes, we conducted electrophysiological recording while the mice were performing the tactile d2-AFC task (Supplementary Fig. 6). We observed consistent age-dependent decreases in the absolute selectivity index for action plan (left/right) (|SI|) and the decoding accuracy of action plan for the delay period (Supplementary Fig. 6), as those observed in calcium imaging. These findings provide further support and

validation for the calcium imaging results, suggesting that the observed signal changes in calcium imaging reflect consistent changes in firing rates across different age groups.

## Age-related changes in crossmodal memory coding

Frontal areas are known to encode task-relevant information across sensory modalities, allowing for modality-general processing[43–46]. To examine how aging affects the crossmodal representation in the mPFC, we tested the mice with a unimodal delayed 2-AFC task, in which a tactile or an auditory stimulus was presented during the sample period in separate trial blocks (Fig. 3a). The mice performed unexpectedly well on the unimodal task even on the first day, except for the compromised performance of the advanced-aged mice in the auditory trials (Fig. 3b), implying that the acquisition of the bimodal-based memory-guided behavior can be easily generalized to each of the component modalities.

Similar to our observation in the bimodal task, we found an age-dependent decrease in the fraction of action-plan-selective delay neurons (Supplementary Fig. 7a, b) as well as in the absolute selectivity index (|SI|) across neurons (Supplementary Fig. 7c, d). Given that fractions of mPFC neurons displayed consistent action-plan selectivity between the tactile and auditory trials (Fig. 4a), we asked how consistently mPFC neurons represented action-plan between the two modalities. First, we computed the correlation between the selectivity indices (SIs) of two modalities (SI correlation) in each imaging session to evaluate the crossmodal consistency of action-plan selectivity. The SI correlation decreased in the middle-aged mice as well as in the advanced-aged mice, compared with the young mice (Fig. 4b). Similarly, SIs multiplied between the two modalities decreased in the middle-aged and advanced-aged groups, in comparison to the young group ($p < 0.001$, one-way ANOVA with Bonferroni post hoc test; Fig. 4c).

We also counted the number of 'consistent' neurons that showed significant SIs with the same sign in the two modalities as in the example neurons shown in Fig. 4a, and found that the fraction of consistent neurons was lower in the middle-aged and advanced-aged groups than in the young group ($U = 398$, $p = 0.0059$ (early delay), $U = 383$, $p = 0.029$ (late delay) for young vs. middle-aged, $U = 364$, $p = 2.6 \times 10^{-4}$ (early delay), $U = 361$, $p = 4.6 \times 10^{-4}$ (late delay) for young vs. advanced-aged, Wilcoxon rank-sum test; Fig. 4d). Next, to quantify the similarity of the temporal dynamics in population activity between the modalities, we computed a similarity index of the state-space trajectories between the two modalities after applying dimensionality reduction to the population activity (Fig. 4e,f; Methods). The similarity index also decreased with age ($p < 0.001$, one-way ANOVA with Bonferroni post hoc test), implicating that the population activity pattern became dissimilar between the two modalities with aging.

To further evaluate the modality-general action-plan coding at the population level, we performed cross-modality decoding of action plan in which a decoder was trained using delay activity of 200 randomly selected neurons for each age group in one modality and tested to classify delay activity in the other modality (Fig. 4g; Methods). The young mPFC exhibited high accuracy of cross-modality decoding with only a small performance drop compared with within-modality decoding, providing the modality-general maintenance of action plan. In contrast, the middle-aged mPFC showed a remarkable decrement in the accuracy of cross-modality decoding compared with within-modality decoding, while the advanced-aged mPFC had low accuracy both in within- and cross-modality decoding. Consequently, the product of multiplying within-modality and cross-modality decoding accuracies decreased with aging (Fig. 4h).

As previously mentioned regarding Fig. 3b, the advanced-aged mice exhibited compromised performance in auditory trials, indicating the likelihood of age-related hearing loss. It is therefore plausible that this hearing loss might disrupt the establishment of auditory-

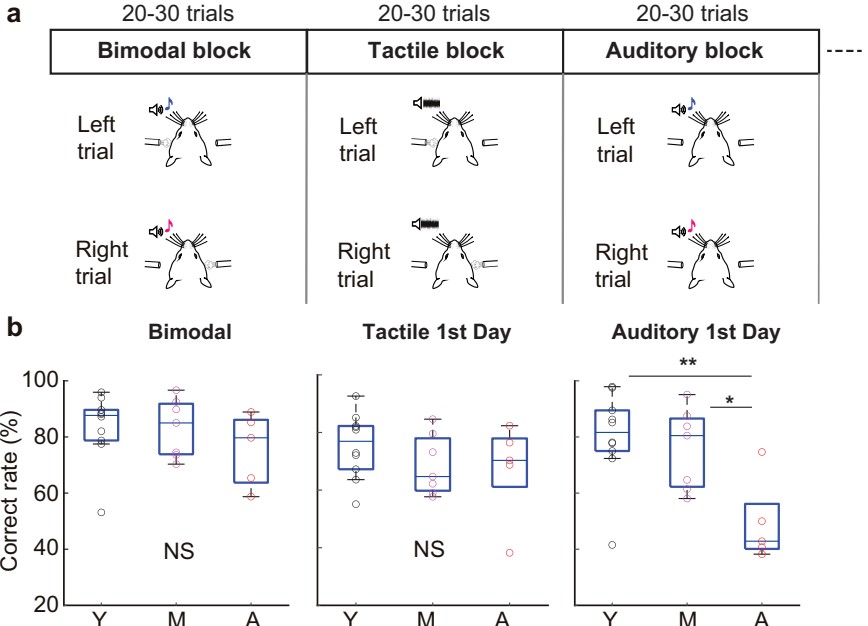

**Fig. 3 | The unimodal delayed 2-AFC task and behavioral performance.**
**a,** Sequence of the unimodal d2-AFC task. Tactile and auditory trial blocks were alternated with the bimodal trial block, and each block was switched every 20–30 trials. Mouse cartoons modified from the drawing[73]. **b** Correct rate on the first day tested with the unimodal d2-AFC task as well as the bimodal task for all the mice

($n = 10$ young mice, $n = 7$ middle-aged mice, $n = 5$ advanced-aged mice). *$U = 60$, $p = 0.02$, **$U = 101$, $p = 0.008$, Wilcoxon rank-sum test. Box plots show the median and the 25th and 75th percentiles as box edges and the whiskers extend to the 5th and 95th percentiles. Y young, M middle age, A advanced age. Source data are provided as a Source Data file.

based representations, thus contributing to the observed reduction in crossmodal representation within the advanced-age group. To address a parallel concern within the middle-aged group, we collected additional imaging sessions, focusing exclusively on sessions where the middle-aged mice exhibited performance comparable to young mice (Supplementary Fig. 8a). Even in these selective sessions, the middle-aged mice exhibited a degradation in modality-general coding when compared to the young mice (Supplementary Fig. 8b–d). This result indicates that the decline in modality-general coding, at least within the middle-aged group, cannot be solely ascribed to hearing loss.

While we have examined the modality-general aspect of action-plan coding, it is also possible that the same population may represent action plan in a modality-dependent manner. To address this, we performed modality-action-plan decoding in which a decoder was trained to classify all four combinations of action plan (left/right) and modality (tactile/auditory) (Fig. 4i). The accuracy of modality-action-plan decoding would be high if the population activity retains modality-dependent action-plan coding. The young and the middle-aged groups showed comparable decoding accuracy ($p > 0.5$, one-way ANOVA with Bonferroni post hoc test; Fig. 4i), which was significantly higher than that in the advanced-aged group ($p < 0.001$, one-way ANOVA with Bonferroni post hoc test). This suggests that modality-dependent action plan coding was retained both in the young and middle-aged groups, but degraded in the advanced-aged group.

Finally, we tested the possibility that any potential variations in licking responses between the modalities and the age groups may have caused the age-dependent change in the crossmodal representation. However, we confirmed that licking responses did not differ among the age groups nor between the modalities ($F(2,38) = 0.94$, $p = 0.40$ for lick rate and $F(2,37) = 0.68$, $p = 0.51$ for inter-lick-interval, the main effect of age, two-way ANOVA with age-group × modality factors; Supplementary Fig. 9a, b). In addition, to further investigate the impact of premature licking during the delay period, we specifically selected trials without premature licking and examined the crossmodal representation. We found that modality-general coding

degraded with aging in the uni-modal task by evaluating the similarity index in state space and cross-modality decoding (Supplementary Fig. 9c–e), both of which replicated the original main results depicted in Fig. 4f–h. In summary, these results revealed functional alterations in the mPFC that already occur in middle age – the young mPFC provided modality-general coding concurrent with modality-dependent coding, whereas modality-general coding was significantly disrupted and only modality-dependent coding was preserved in the middle-aged mPFC. Both types of coding then diminished in the advanced-aged mPFC.

## De-differentiation of the mPFC circuit

Computational modeling and theory posit that attractor dynamics implemented by the recurrent networks sustain the delay activity even after the cessation of sensory input[47–50]. We hypothesized that the action-plan-selective delay neurons, especially those sharing the same selectivity, are mutually connected for robust maintenance of delay activity (Fig. 5a) and that the weakened connectivity could lead to the age-associated decline in the action-plan signal.

To test this hypothesis, we measured spontaneous activity during the resting state before the mice performed the bimodal d2-AFC task and computed pairwise correlation between neurons based on the binarized calcium events to evaluate resting-state functional connectivity (RSFC). As predicted, in the young mice the RSFC of action-plan-selective delay neurons with the same sign of SI (termed 'task-relevant' RSFC) was higher than 'non-task-relevant' RSFC (i.e., RSFC of action-plan-selective delay neurons and non-task-related neurons) across inter-neuronal distances (Fig. 5b, c). Notably, the difference between task-relevant and non-task-relevant RSFC diminished as a function of age (Fig. 5c).

The age-dependent variation in RSFC cannot be accounted for by spontaneous licking responses during the resting state since there were no substantial differences among the age groups (Supplementary Fig. 10a, b). Alternatively, possible differences in the spontaneous activity level between action-plan-selective delay neurons and non-

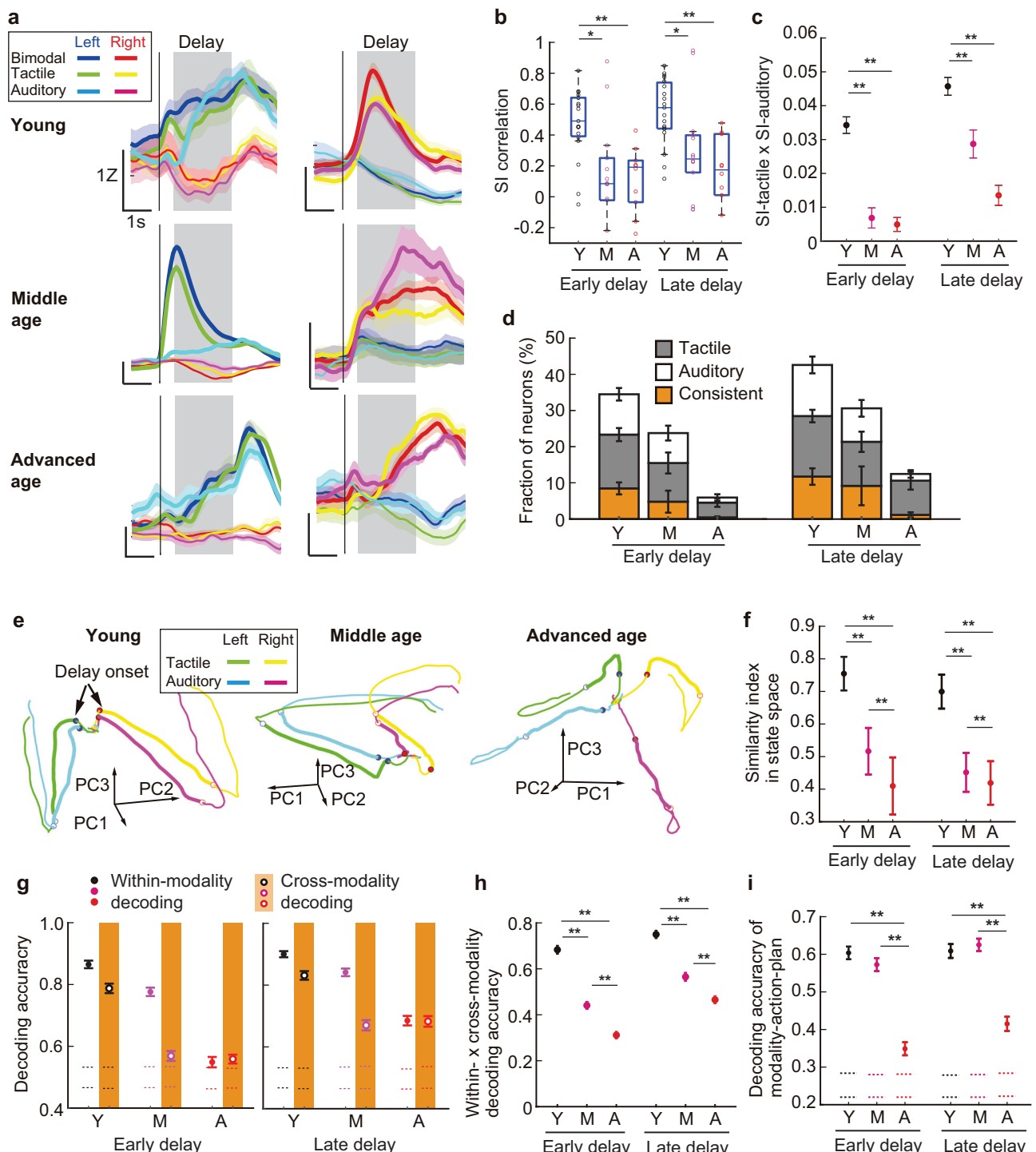

**Fig. 4 | Modality-general coding degrades with aging. a** Trial-averaged activity of the example consistent action-plan-selective delay neurons in the unimodal task. Gray shading, delay; vertical line, sample onset. Colored shadings, ± s.e.m. **b** Correlation of selectivity index (SI) between tactile and auditory trials for the imaging sessions (young, $n = 19$ sessions, 10 mice; middle age, $n = 14$ sessions, 7 mice; advanced age, $n = 10$ sessions, 5 mice). Box plots, the median and the 25th and 75th percentiles as box edges with the whiskers extending to the 5th and 95th percentiles. $*U = 401$, $p = 0.0048$, $**U = 359$, $p = 7.5 \times 10^{-4}$ for early delay, $*U = 397$, $p = 0.0074$, $**U = 362$, $p = 4.5 \times 10^{-4}$ for late delay, Wilcoxon rank-sum test. Y, young; M, middle age; A, advanced age. **c**, SI in tactile × SI in auditory trials averaged across all neurons (young, $n = 1,577$ neurons; middle age, $n = 689$ neurons; advanced age, $n = 606$ neurons). Error bars, ± s.e.m. $**p < 0.001$, one-way ANOVA with Bonferroni post hoc test. **d** Fraction of consistent, tactile, and auditory action-plan-selective

delay neurons (Methods) (young, $n = 19$ sessions, 10 mice; middle age, $n = 14$ sessions, 7 mice; advanced age, $n = 10$ sessions, 5 mice). Error bars, ± s.e.m. **e** Example trajectories of population activity in state space (only 3 PC axes shown for display purposes, Methods). Filled circle, delay onset; open circle, delay end. **f** Similarity index in state space. $**p < 0.001$, one-way ANOVA with Bonferroni post hoc test. Error bars, ± s.d. across 1000 iterations. **g** Within- and cross-modality decoding of action plan (Methods). Horizontal lines, 97.5th and 2.5th percentile of the shuffled distribution. Error bars, ± s.e.m. across 100 iterations. **h** Product of multiplying within-modality and cross-modality decoding accuracy. $**p < 0.001$, one-way ANOVA with Bonferroni post hoc test. Error bars, ± s.e.m. across 100 iterations. **i** Modality-action-plan decoding (Methods). $**p < 0.001$, one-way ANOVA with Bonferroni post hoc test. Error bars, ± s.e.m. across 100 iterations. Source data are provided as a Source Data file.

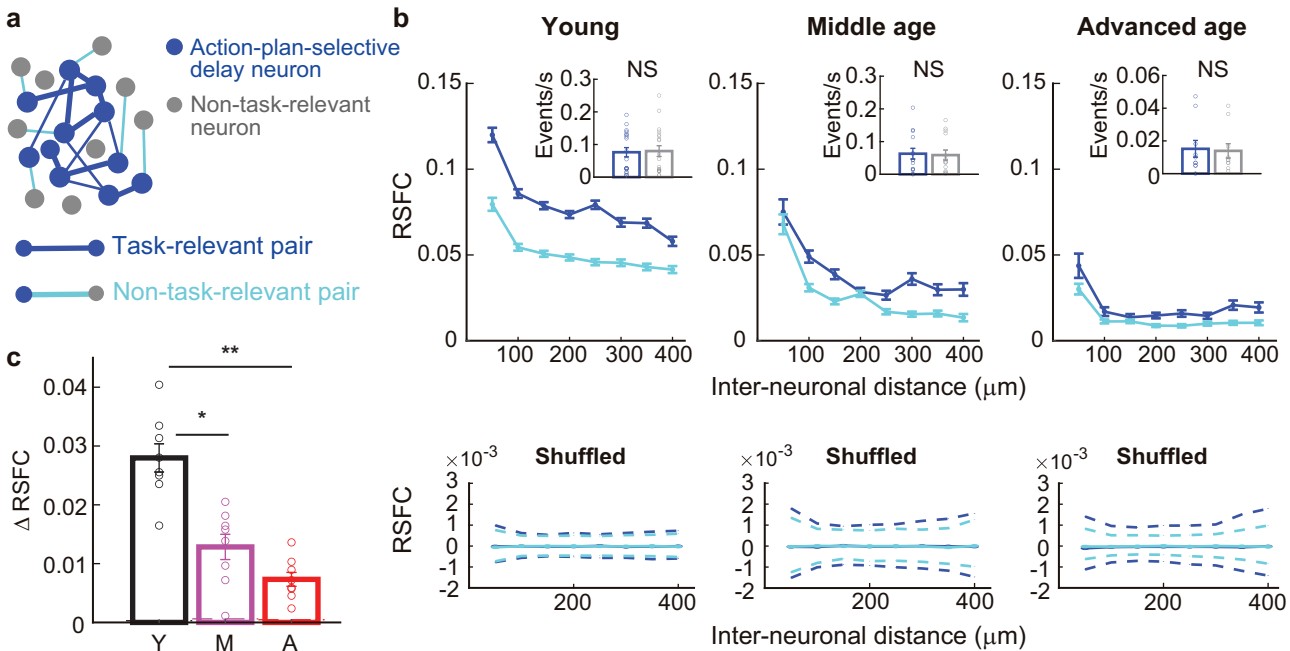

**Fig. 5 | Resting-state functional connectivity (RSFC) degrades with aging.**
**a** Schematic of the functional connectivity of mPFC neurons. **b** Top, RSFC as a function of the inter-neuronal distance. Task-relevant RSFC (blue lines) was computed between pairs of action-plan-selective delay neurons that had the same sign of action-plan selectivity for each imaging session (young, $n = 17,427$ pairs from 19 sessions; middle-age, $n = 4506$ pairs from 14 sessions; advanced-age, $n = 5277$ pairs from 10 sessions). Non-task-relevant RSFC (light-blue lines) was computed between action-plan-selective delay neurons and non-task-relevant neurons (young, $n = 23,787$ pairs from 20 sessions; middle age, $n = 8175$ pairs from 14 sessions; advanced age, $n = 17,093$ pairs from 10 sessions). Error bars denote ± s.e.m. Inset, the average calcium event rate during the resting state was not different between action-plan-selective delay neurons (blue) and non-task-relevant neurons (gray) for any of the age groups ($U = 379$, $p = 0.82$ for young, $U = 197$, $p = 0.80$ for

middle age, $U = 117$, $p = 0.39$ for advanced age; Wilcoxon rank-sum test). Young, $n = 1072$ action-plan-selective delay neurons and 394 non-task-relevant neurons, 19 sessions; middle age, $n = 408$ action-plan-selective delay neurons and 258 non-task-relevant neurons, 14 sessions; advanced age, $n = 444$ action-plan-selective delay neurons and 417 non-task-relevant neurons, 10 sessions. Bottom, same as top, but after shuffling the calcium events across the time bins. Blue and light-blue lines, median of the shuffled distribution for the task-relevant and non-task-relevant RSFC, respectively. Dashed lines, 2.5th and 97.5th percentile. Note the smaller scale compared with the top panels. **c** Difference between the task-relevant and non-task-relevant RSFC. Open circles, each inter-neuronal distance bin. *$p < 0.02$, **$p < 0.0005$, one-way ANOVA with Bonferroni post hoc test. Error bars denote ± s.e.m. Y, young; M, middle age; A, advanced age. Source data are provided as a Source Data file.

task-related neurons might have generated the difference in RSFC. However, the mean calcium event rate was not significantly different between action-selective delay neurons and non-task-related neurons in all age groups (Fig. 5b, insets). Moreover, we shuffled the calcium events among the time bins thereby disrupting the temporal structure of spontaneous activity while preserving the average activity level. The shuffling eliminated the age-dependent difference between the task-relevant RSFC and non-task-relevant RSFC ($F_{(2,21)} = 1.74$, $p = 0.20$, one-way ANOVA), confirming that the age-dependent effect on RSFC cannot be attributed to a mere difference in the activity level between the two populations. These results support our hypothesis that with aging, the connectivity of recurrent circuits in the mPFC is weakened and less differentiated for memory maintenance.

The results of RSFC revealed the age-dependent decline of the task-relevant network even in the absence of task engagement. To extend our understanding, we investigated the functional connectivity (FC) in the task context by computing the pairwise correlations between action-plan-selective delay neurons based on the activity during the delay period and the baseline period in the bimodal task. The FC both for the delay and the baseline periods declined as a function of age ($F_{(2,37)} = 85.0$, $p < 1.0 \times 10^{-5}$, the main effect of age group, three-way ANOVA with age group × task period × interneuronal distance; Supplementary Fig. 11a), consistent with the RSFC shown in Fig. 5. To test the FC's importance for behavioral performance, FC was compared between correct trials and error trials. The FC during the delay period for action-plan-selective delay neurons was significantly larger in correct trials than in error trials only in the young group

($F_{(1,7)} = 16.5$, $p = 0.0048$, the main effect of behavioral performance, two-way ANOVA with behavioral performance × interneuronal distance), but not in other age groups ($F_{(1,7)} = 0.93$, $p = 0.37$ for middle age, $F_{(1,7)} = 0.50$, $p = 0.50$ for advanced age; Supplementary Fig. 11b). The FC during the baseline period was also significantly large in correct trials than in error trials only in the young group ($F_{(1,7)} = 11.2$, $p = 0.012$), but not in other age groups ($F_{(1,7)} = 0.04$, $p = 0.84$ for middle age, $F_{(1,7)} = 2.56$, $p = 0.15$ for advanced age; Supplementary Fig. 11c). These results suggest that the FC between action-plan-selective delay neurons both during the baseline and delay periods was predictive of the upcoming performance of working memory in the young group, but such predictive functional coupling is no longer detectable in middle-aged and advanced-aged groups. Together with the results of RSFC shown in Fig. 5, these findings revealed the age-dependent decline in the task-relevant network both during task engagement and outside of the task context.

Next, we examined the FC in the unimodal task. We classified neurons as 'bimodal' if they exhibited a significant and the same sign of SI during the delay period in both tactile and auditory trials. Neurons that showed significant SI only in either the tactile or auditory trials, and did not belong to the bimodal category, were referred to as 'tactile-unimodal' or 'auditory-unimodal' neurons, respectively. To assess the FC between unimodal neurons, we calculated the FC between pairs of tactile-unimodal neurons and pairs of auditory-unimodal neurons. In the delay period, bimodal neurons exhibited significantly higher FC compared to unimodal neurons in the young group ($F_{(1,7)} = 12.2$, $p = 0.010$, the main effect of neuron type, two-way ANOVA with

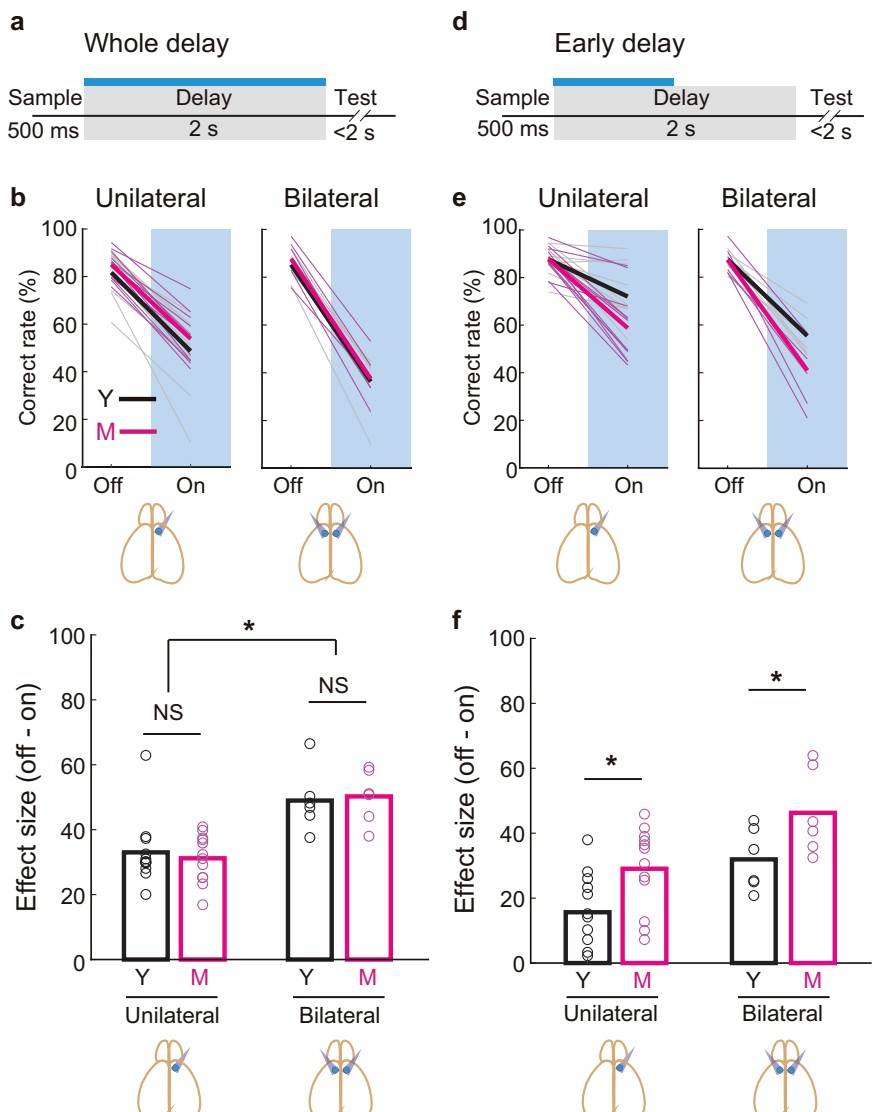

**Fig. 6 | Optogenetic inactivation of the mPFC reveals age-dependent vulnerability to perturbations. a** Optogenetic inactivation during the whole delay period. Pyramidal neurons in the mPFC expressing stGtACR2 were inactivated with blue light. **b** Left, unilateral inactivation. The correct rate decreased both in young (young, $n = 6$ mice × 2 hemispheres, black lines) and middle-aged mice (middle age, $n = 6$ mice × 2 hemispheres, magenta lines) ($p < 0.0001$ both for young and middle-aged, one-sided bootstrap, Methods). Right, bilateral inactivation. The correct rate decreased both in young ($n = 6$ mice) and middle-aged mice ($n = 6$ mice) ($p < 0.0001$ both for young and middle-aged mice). Faint colored lines, individual mice. **c** Behavioral effect size measured as the difference in the correct rate between laser-off and laser-on conditions. Bilateral inactivation produced a larger effect size than unilateral inactivation in both age groups ($p < 0.0001$ both for

young and middle-aged mice, one-sided bootstrap). The effect size was not different between the two age groups in either unilateral ($p = 0.11$, one-sided bootstrap) or bilateral inactivation ($p = 0.29$, one-sided bootstrap). **d** Optogenetic inactivation during the early delay period. **e** Both unilateral and bilateral inactivation decreased the correct rate both in young (black lines) and middle-aged mice (magenta lines) ($p < 0.0001$ for all comparisons). **f** Behavioral effect size was significantly larger in the middle-aged mice than the young mice both in unilateral ($p < 0.0001$, one-sided bootstrap) and bilateral inactivation ($p < 0.0001$, one-sided bootstrap). Bilateral inactivation produced a larger effect size than unilateral inactivation in both age groups ($p < 0.0001$ for both of the age groups, one-sided bootstrap). Mouse brain cartoons (**b**, **c**, **e**, **f**, bottom) modified from the drawing[73]. Y, young; M, middle age. Source data are provided as a Source Data file.

meuron type × interneuronal distance). This effect was weakly observed in the middle age group ($F(1,7) = 6.15$, $p = 0.042$), but not significant in the advanced age group ($F(1,7) = 0.59$, $p = 0.47$) (Supplementary Fig. 12a). In the baseline period, bimodal neurons demonstrated significantly higher FC than unimodal neurons in young adults ($F(1,7) = 9.9$, $p = 0.016$), but not in the other age groups ($F(1,7) = 0.18$, $p = 0.68$ for middle age, $F(1,7) = 0.30$, $p = 0.60$ for advanced age; Supplementary Fig. 12b). These findings suggest that bimodal neurons exhibit stronger functional coupling with each other, forming a more robust functional network compared to unimodal neurons in young adults. However, the strength of the bimodal neuron network deteriorates with age.

## Age-dependent vulnerability to perturbation

Although our results have shown that several aspects of degradations in the mPFC circuit are manifested already in middle age, the middle-aged mice did not display an apparent behavioral decline compared with the young mice after learning (Figs. 1c, 3b, and Supplementary Fig. 1c). We reasoned that the middle-aged mPFC circuit, despite being degraded, was nevertheless able to support task performance in the normal condition, but it should have an intrinsic vulnerability to perturbations.

To test this possibility, we performed optogenetic inactivation of mPFC pyramidal neurons expressing inhibitory opsin (stGtACR2)[51] in the delay period of the bimodal task (Fig. 6a and Supplementary

Fig. 13). The bilateral inactivation during the whole delay period severely impaired the behavioral performance of both age groups, demonstrating that the delay activity of the mPFC is necessary for memory maintenance ($p < 0.0001$, one-sided bootstrap; Fig. 6b). The sham laser stimulation with blocking laser pathway confirmed that potential light leak did not affect the task performance both in the young and the middle-aged mice ($p = 0.12$, 0.27 for young and the middle-aged, one-sided bootstrap; Supplementary Fig. 14a–c). Unilateral inactivation also caused impairment although the effect size was smaller than bilateral inactivation ($p < 0.0001$, one-sided bootstrap; Fig. 6c). The effect size was not different between the age groups ($p = 0.11$, 0.29 for the unilateral and bilateral inactivation, one-sided bootstrap; Fig. 6c). In contrast, a brief inactivation applied only during the early delay period (Fig. 6d) produced a differential effect size between the age groups: the middle-aged mice showed more severe impairment than the young mice both in the unilateral and the bilateral inactivation ($p < 0.0001$, one-sided bootstrap; Fig. 6e, f). These results hence revealed an age-dependent vulnerability of the mPFC function to a brief perturbation.

## Discussion

Using calcium imaging and optogenetic manipulation, we were able to demonstrate that several aspects of the functional organization of the mPFC circuit change with aging. First, we found a significant age-related reduction both in the fraction of action-plan selective delay neurons and in the action-plan signal encoded by individual neurons, thereby degrading decodability of the maintained memory in the mPFC. This finding in the mouse mPFC is in line with a previous electrophysiology study in monkeys that also showed an age-related decrease in the spatial memory selectivity coded in dorsolateral PFC neurons during an oculomotor delayed response task[39]. This convergent evidence points to a common effect of aging on the memory maintenance mechanism between rodents and primates.

We observed that the mice exhibited premature licking during the delay period, which predicted the direction of licking during the subsequent test period. This raises the possibility that motor act, in addition to working memory, could influence behavioral performance and age-dependent activity. To address this possibility, we first confirmed that the occurrence of premature licking was not age-dependent. Furthermore, even when considering only trials without premature licking, we consistently observed the age-related decline in the action-plan signal during the delay period. These findings provide compelling evidence that although a premature licking bias was present during the delay period, it did not account for the age-dependent decline in the action-plan selectivity.

Our results further revealed an age-dependent alteration in the crossmodal memory representation. In the young mPFC, population activity provided modality-general coding for action plan, in line with the general properties of frontal areas[43–46,52], and concurrently, they also allowed the modality-dependent representation in the same population. The modality-general coding was remarkably disrupted in the middle-aged mPFC while preserving modality-dependent coding, and both types of coding were retarded in the advanced-aged mPFC. The retarded modality-general coding in the advanced-aged group could be a result of a natural progression, in light of their compromised auditory-based performance illustrated in Fig. 3b. The presence of age-associated hearing loss likely disrupted the formation of auditory-based representations, potentially contributing to the subsequent reduction in both crossmodal and modality-dependent representations. Future investigations incorporating unimodal 2-AFC tasks without delay, alongside the d2-AFC tasks, will help discern whether any behavioral impairment in different aging stages stems from perceptual disabilities or working memory deficits. However, an unexpected revelation occurred when middle-aged mice displayed a decline in modality-general coding, despite showcasing auditory-based performance comparable to that of young mice (Supplementary Fig. 8). This unforeseen outcome prompts the suggestion that the dip in modality-general coding, at least within the middle-age phase, represents an age-dependent process that extends beyond the realm of hearing loss. Our findings highlight the age-dependent degradation of crossmodal memory maintenance in the mPFC. This can affect the ability of the prefrontal circuits to integrate crossmodal information and regulate goal-directed behavior across different sensory modalities[43,44,52].

The comparable performance between young and middle-aged mice implies that the degradation of modality-general coding in the mPFC may precede the decline in working memory performance associated with aging. This raises a pivotal question: what implications do alterations in mPFC neural activity during aging hold? We posit that beneath overt behavioral performance, the underlying neural computations are trending towards suboptimal functioning and reduced resilience to perturbation. These trends manifest in observed alterations in mPFC neural activity and functional connectivity. Indeed, as illustrated through the optogenetic manipulation in Fig. 6d–f, the mPFC circuit of the middle-aged group exhibits heightened vulnerability to perturbation, despite seemingly normal behavioral performance in the baseline condition. This underscores that the apparent behavioral performance conceals an emerging fragility in the neural processes supporting it, hinting at a potential susceptibility to disruptions in middle-aged individuals.

The RSFC analysis revealed that the age-related decline of the action-plan representation could be due to the weakening of functional connectivity in the mPFC. As computational modeling studies suggested[47–50], a recurrent attractor network can robustly sustain the selective information in the local circuit. In line with this and as proposed by a review[53], we found that the action-plan selective delay neurons sharing the same direction selectivity exhibited higher functional connectivity compared with that of other irrelevant pairs in the young mPFC. Moreover, our results showed that aging reduced the difference in functional connectivity between the task-relevant pairs and non-task-relevant pairs. A similar age-associated 'de-differentiation' of the human brain-wide networks has been reported in the resting-state functional magnetic resonance imaging (fMRI) studies[4,54,55]. In young adults, different brain-wide networks such as the default mode, dorsal attentional, and salience networks, show distinct connectivity profiles across brain areas, but the distinctness among these networks is reduced in older adults. Our finding implies that de-differentiation can occur also at the local circuit level in the mPFC as early as middle age.

Consistent with the RSFC findings, we also found an age-associated decrease in the functional connectivity (FC) between action-plan-selective delay neurons during task engagement. Notably, in the young group, the FC during both the delay and baseline periods was significantly higher in correct trials compared to error trials. However, this differential FC, which was predictive of behavioral performance, was no longer detectable in the middle-aged and advanced-aged groups. These results indicate that FC was initially predictive of working memory performance but deteriorated with age. In the context of the unimodal task, bimodal neurons, which exhibited significant and consistent selectivity in both tactile and auditory trials during the delay period, demonstrated significantly higher FC compared to unimodal neurons in the young group. This effect was weakly observed in the middle-aged group but not significant in the advanced-aged group. These findings indicate that bimodal neurons form a more robust functional network compared to unimodal neurons in young adults. However, the strength of this network deteriorates with age. The deterioration of functional connectivity among bimodal neurons suggests a disruption in coordinated neural activity associated with modality-general working memory processes.

The age-related deterioration of functional connectivity could be associated with the detectable morphological changes in pyramidal neurons as already reported in several cortical areas[36–38]. Histological examinations of the aged brain have shown that spine densities were reduced with aging in several cortical areas including the prefrontal cortex. In particular, the synapse density and the overall spine size of pyramidal neurons in the monkey dorsolateral PFC were predictive of the animals' learning efficiency of a working memory task[37]. Given that functional connectivity in the resting state agrees with structural connectivity[56], the age-related deterioration of functional connectivity in the present study can likely be accompanied by possible declines in the spine density of pyramidal neurons in the mPFC. It is also important to note that not only the local circuit in the PFC, but the long-range circuits across areas are important for memory maintenance[6,20,24,26,35,57–60]. Indeed, cognitive operation including working memory requires dynamic interareal communications by consistently routing the PFC[6,12,31,61], which is significantly retarded in aging[33,35]. Further studies will be required to investigate the relationship between cognitive aging and brain-wide communications.

The optogenetic inactivation during the delay period confirmed the necessity of the mPFC for action-plan maintenance. Bilateral inactivation caused more severe impairment than unilateral inactivation, which is consistent with a previous study[18]. In line with this, our study also suggests that the action-plan information diminished by optogenetic perturbations in one hemisphere can be partially restored by information from the other hemisphere, whereas bilateral perturbations disrupt the restoration, resulting in a more severe behavioral deficit. Our results with brief inactivation during the early delay further revealed the age-dependent susceptibility of the memory-maintenance circuit to perturbations. In summary, our findings provide insights into the age-dependent alterations of the PFC circuit for memory maintenance – which is already manifested in middle age.

## Methods

### Animals
All experimental procedures were approved by the Animal Care and Use Committee (IACUC) at Nanyang Technological University (protocol#A19062). Calcium imaging experiments were performed in CaMKIIα-Cre mice (Jackson lab stock 005359). Age groups were defined as[62], young (3–8 months; $n = 11$ mice (6 males, 5 females)), middle age (11–14 months; $n = 7$ mice (3 males, 4 females)), and advanced age (18–23 months; $n = 5$ mice (2 males, 3 females)). They were used for calcium imaging in the bimodal delayed two-alternative forced choice (d2-AFC) task. Among them, 10 young, 7 middle-aged, and 5 advanced-aged mice were further used for calcium imaging in the unimodal d2-AFC task; and 9 young, 7 middle-aged, and 5 advanced-aged mice were used for calcium imaging during the resting state. In addition, for the variable reward-size experiment, 5 young (2 males, 3 females), 5 middle-aged (2 males, 3 females), and 4 advanced-aged mice (2 males, 2 females) were used. For the electrophysiology experiments, 2 young (1 male, 1 female), 2 middle-aged (2 males), and 2 advanced-aged mice (1 male, 1 female), were used. For optogenetic inactivation of pyramidal cells, CaMKIIα-Cre mice injected with AAV1-hSyn1-SIO-stGtACR2-FusionRed were used including young (4–7 months; $n = 6$ mice (3 males, 3 females)) and middle-aged mice (11–14 months; $n = 6$ mice (3 males, 3 females)). Animals were housed on a 12 h dark/12 h light cycle (light on between 7:00 and 19:00) at around 21 °C and 62% humidity. All the experiments were carried out between 8:00 and 17:00.

### Surgery
Adult mice were anesthetized with isofluorane (3% for induction and 1.5–2% for maintenance in oxygen) and placed on a stereotaxic frame (RWD Life Science). The mice received buprenorphine (0.05 mg/kg)

before and after surgery and supplementary analgesia (meloxicam, 5 mg/kg) after surgery.

For imaging experiments, animals underwent the following surgical procedures. First, a craniotomy of ~500 μm was made to target the mPFC (AP + 1.8–2.1 mm, ML 0.5 mm from bregma, DV 0.6–1.2 mm from pia surface, the right hemisphere), where we injected 600–800 nL of AAV1-flex-GCaMP6f (100833-AAV1, Addgene, titer $1.2 \times 10^{13}$ vg/mL, diluted 3 times) through a borosilicate glass pipette using a microinjector (Nanoliter 2010, WPI). After a week of recovery, we performed a second surgical procedure to implant the cuffed gradient refractive index (GRIN) lens (Inscopix, diameter: 1 mm; length: 4.2 mm; pitch: 0.5; numerical aperture: 0.5) to target the mPFC (center: AP + 1.8–2.1 mm, ML 0.5 mm, DV 0.6–1.2 mm from pia surface, the right hemisphere) along with a stainless steel headplate affixed to the skull using machine screws and black dental cement (Contemporary Ortho-Jet, Black, Lang Dental). We expanded the virus-injection craniotomy to ~1 mm and slowly aspirated the superficial tissue using a blunt 27 G needle connected to a vacuum pump, avoiding blood clotting by constantly irrigating the tissue with sterile Ringer's solution. After ensuring there was no active bleeding, the cuffed GRIN lens was gently placed upon the tissue and the lens was fixed to the skull with super glue (Scotch super glue AD125, 3 M). The GRIN lens was then cemented to the rest of the implant and the surface of the cuffed lens was covered with a silicone elastomer (Kwik-Cast). After a week of recovery, the mice underwent behavioral training. Once the mice reached criterion performance (see "Behavioral procedure" below), we implanted the baseplate for chronic imaging[63] under 1–1.5% isoflurane anesthesia. Silicone sealant was removed to expose the surface of the lens and the miniature microscope attached to the baseplate was lowered with a micromanipulator to the desired focal plane. The baseplate was then cemented to the rest of the implant, and covered with a protective cap after retraction of the microscope.

For electrophysiology experiments, animals underwent the headplate implantation surgery, which involved marking the surface of the skull above the mPFC (center: AP + 1.8–2.1 mm, ML 0.5 mm) and implanting a stainless steel headplate affixed to the skull using machine screws and black dental cement.

For optogenetic inactivation experiments, we injected AAV for stGtACR2[51] (AAV1-hSyn1-SIO-stGtACR2-FusionRed, 105677-AAV1, Addgene, titer $1.9 \times 10^{13}$ vg/mL, diluted 5 times, 500–600 nL) in the bilateral mPFC (AP + 1.8–2.1 mm, ML 0.4–0.5 mm, DV 0.8–1.0 mm) of CaMKIIα-Cre mice. After a week of recovery, we implanted a headplate and fiber optic cannula (Ø1.25 mm ceramic ferrule, Ø400 μm Core, 0.39 NA; Thorlabs) bilaterally to deliver light to the mPFC.

### Histology
We performed histology to confirm the location of the implanted GRIN lens in imaging experiments and the optic fiber used in optical stimulation experiments. At the end of the experiments, the mice were deeply anesthetized with isoflurane and immediately perfused with chilled 0.1 M PBS followed by 4% paraformaldehyde (wt/vol) in PBS. The brain was removed and post-fixed overnight at 4 °C. After fixation, the brain was placed in 30% sucrose (wt/vol) in PBS solution at least for 1–2 days at 4 °C. After embedding and freezing, the brain was sectioned into 50 μm coronal slices using a cryostat (Leica CM1950). The slides were mounted with Fluoroshield with DAPI (Sigma-Aldrich) and examined with a Microphot-SA fluorescence microscope (Nikon Corp.).

### Behavioral procedure
**Bimodal delayed two-alternative forced choice (d2-AFC) task training.** Behavioral training for head-fixed mice was conducted through a set of customized codes within the Presentation software (Neurobehavioral Systems). Habituation (2–3 days) was conducted

first, where no stimulus was presented, and the mice were given free water rewards ( ~2–3 μL) for each lick on either of two metal lickports placed to the left and the right sides. Licks were detected by each of the two lickports via an electrically coupled circuit board. The conditioning phase was executed next, in which the mice were trained to lick either left or right in response to a bimodal stimulus including tactile stimuli (air puffs targeted to either the left or right cheek) and an auditory stimulus (either a 2 kHz or 8 kHz tone, ~75 dB). An airpuff given to the left cheek together with a 2 kHz tone was referred to as a 'Left' trial, and an air puff to the right cheek with an 8 kHz tone as a 'Right' trial. After the stimuli were given, a screen lit up for 2 s, representing the response window that accepted licking responses. During the conditioning phase, there was only a short delay (100 ms) between the end of the stimulus and the start of the response window. Licking left in 'Left' trials within the response window was rewarded (Correct), and licking right was punished (Commission error) with a beep sound and a longer inter-trial interval (additional 7 s plus the default 7 s); vice versa for 'Right' trials. If no lick was detected during the response window (Omission error), water reward was given after the response window ended. In the training phase, the delay between the end of the stimulus and the start of the response window was extended incrementally by 200–300 ms whenever the mice reached ~70% correct rate in 20–30 trials. The correct rate was calculated as the percentage of Correct trials divided by all trials.

**Unimodal delayed two-alternative forced choice (d2-AFC) task.** After training mice with the bimodal d2-AFC task with a 2 s delay period and collecting the calcium imaging data, the mice were further tested with the unimodal d2-AFC task under calcium imaging. In the unimodal d2-AFC task, auditory, tactile, or bimodal stimulus was presented as an interleaving block with each block consisting of 20–30 trials (Fig. 3a). In all blocks, the trial structure was the same as the bimodal d2-AFC task except the presented stimulus during the sample period: in the auditory trial block, only the auditory stimulus was presented (2 kHz or 8 kHz for the left or the right trial, respectively), and in the tactile trial block the airpuff stimulus (given to left or right cheek) was presented along with a white noise sound from the speaker. In the bimodal block, the presented stimulus was the same as the bimodal d2-AFC task. The sequence of the blocks – bimodal, tactile, auditory, bimodal, auditory, tactile – was repeated until mice were sated.

To evaluate how reward size impacts behavior and neural activity across age groups, we conducted training on a tactile d2-AFC task using a different set of mice. This tactile d2-AFC task was identical to the bimodal task except that only the airpuff stimulus (applied to left or right cheek) was presented during the sample period. This training procedure was the same as that of the bimodal task described above. Once the mice reached an approximate 70% correct rate with a 2-s delay, we introduced two different reward size conditions (default and high reward conditions). In the default reward condition, the reward size was the same as the one employed during training. In separate sessions, we tested the high reward condition wherein the reward size was doubled compared to the default condition.

In the electrophysiology experiments, mice underwent training using the tactile d2-AFC task in which only the airpuff stimulus was presented during the sample period. The duration of the delay period was gradually extended, and recording commenced once the mice achieved an approximate 70% correct rate with a 1.5-s delay.

**Microendoscopic calcium imaging**
The detailed procedure was described elsewhere[21,42]. Briefly, we performed cellular-resolution microendoscopic calcium imaging from genetically defined cell types in the right mPFC using a miniaturized integrated fluorescence microscope (Inscopix; 20x objective; LED power: 0.2–0.7 mW; CMOS sensor resolution: 1,200 × 800 pixels)

coupled to a GRIN lens[63]. Images were acquired at 15–20 frames/s and spatially downsampled by a factor of 2 using nVista3 (Inscopix). Behavioral events were registered offline with imaging frames by acquiring analogue voltage output from both the nVista3 system and the behavioral control system via NI PCIe-6321 with a custom code in LabVIEW (National Instruments).

**Electrophysiology**
For electrophysiology recording from behaving mice, a craniotomy ( ~1 mm diameter) was made over the target area in the mPFC (center AP + 1.8–2.1 mm, ML 0.3–0.5 mm) under gas anesthesia (1.5–2% isoflurane in oxygen). Mice were allowed to recover at least 4 h before the recording. Mice were then placed in a tube and their head was fixed for the task performance. A silicon probe (ASSY-37 H8b, Acute 32 channel H8b probe, 1 shank, 30 μm site spacing, 9 mm length, Cambridge NeuroTech) was inserted at a depth range of 1000 to 1200 micrometers using a motorized micromanipulator (MM-500-2, RWD Life Science). Wideband neural signals were sampled at 30 kHz with RHX data acquisition software in the Intan 128-channel acquisition system (M4200, Intan Technologies). After every daily experiment, the craniotomy was sealed with a silicone elastomer (Kwik-Cast, World Precision Instruments).

**Optogenetic inactivation**
For optogenetic inactivation, a blue laser was delivered through a patch cable connected with DPSS laser (473 nm; Shanghai laser) under the control of a custom-made LabVIEW software (30 Hz pulse trains, 0.7–0.9 mW at fiber tip). The patch cable was connected with the ferrule end of optic fiber cannula (implanted in the brain) via a mating ceramic sleeve (ThorLabs). Laser stimulation was applied in 20% of randomly selected trials. In the whole-delay condition, laser stimulation started with the delay period onset and continued for 2 s to cover the whole delay period window in each trial. In the early delay condition, laser stimulation started with the delay period onset and lasted for 1 s, thus covering the early portion of the delay period in each trial. For the sham optogenetic experiment, a black vinyl tape was placed at the junction between the patch cable and the ferrule end of optic fiber cannula to block the laser pathway while the other laser stimulation parameters were kept the same as the whole delay condition.

**Imaging data analysis**
Image stacks were first corrected for small displacements of the brain by registering to the first frame of the image sequences[63]. To remove scattered fluorescence and background neuropil signal, a spatial Gaussian high-pass filter was applied ($\sigma = 50$ μm) and the relative changes in fluorescence:

$$dF(t)/F0 = (F(t) - F0)/F0 \qquad (1)$$

were computed for each pixel, where $F0$ indicates the mean activity of each pixel during the entire session and $dF(t)$ is the mean subtracted fluorescence at each time point $t$. The successive application of principal component and independent component analyses (PCA/ICA algorithm) was performed for the $dF(t)/F0$ videos to extract the activity of individual neurons[64,65]. ROIs that have signal-to-noise ratio of > 3 and only one component from the output of the PCA/ICA algorithm were accepted as real neurons. To correct for decreases in baseline fluorescence due to bleaching of the calcium indicator we subtracted slow fluctuations in baseline according to the expression:

$$F_{corrected}(t) = F(t) - G(t) + <F(t)>, \qquad (2)$$

where $F(t)$ represents the output trace obtained from the PCA/ICA algorithm, brackets indicate time average over the entire recording session, and $G(t)$ is the[6] average of $F(t)$ over a 300-s sliding window.

Finally, the corrected calcium traces $F_{corrected}(t)$ were z-scored using whole traces in each ROI. Unless otherwise stated, we used the z-scored activity for all the analyses.

## Action plan selectivity

To evaluate how each neuron showed activity biased to either of the action direction (Left or Right), we computed the selectivity index (SI) for action plan by comparing activity during each of the following task periods: baseline (1000 ms before the sample onset), sample (0–500 ms from the sample onset), early delay (0–1000 ms from the delay onset), late delay (1000–2000 ms from the delay onset), and feedback (0–2000 ms from the first lick after the test onset). Using only correct trials, the trial-by-trial activity in left and right trials was collected to construct a Receiver Operating Characteristic (ROC) curve. The area under the ROC curve (AUC) is the probability that a randomly sampled activity for left trials is larger than a randomly sampled activity for right trials. The SI was computed as:

$$SI = 2 \times (AUC - 0.5) \qquad (3)$$

such that ±1 represents perfect discriminability and 0 indicates chance-level discriminability. Positive and negative SI denotes a preference for left and right trials respectively. To determine its statistical significance, we randomly relabeled each trial as left or right and computed SI in each permutation. Each permutation was repeated 1000 times to obtain the null distribution. The *p*-value of the observed SI was calculated as a percentile of the null distribution. In the unimodal d2-AFC task, we categorized distinct types of action-plan-selective delay neurons based on SI (Fig. 4d). Neurons exhibiting the same sign of significant SI in both tactile and auditory trials were labeled as 'consistent' neurons. 'Tactile' and 'auditory' neurons were identified as action-plan-selective delay neurons during tactile and auditory trials, respectively, with the exclusion of consistent neurons.

## Decoding analysis

Single-trial decoding for left vs. right trials was performed on a pseudo-population assembled across sessions. To estimate the variance of the decoding performance, on each iteration of the decoder (100 iterations), we randomly selected 300 neurons (for the bimodal d2-AFC) or 200 neurons (for the unimodal d2-AFC) in each age group. We matched the number of trials per trial type (15 trials each for left and right trials in the bimodal d2-AFC; 10 trials each for left/right with tactile/auditory modalities in the unimodal d2-AFC) contributed by each neuron that was selected to participate in the population decoding. Using z-scored activity for each neuron, we performed 5-fold cross-validation with a linear support vector machine (SVM) decoder to estimate the classification performance ('fitcecoc' function in MATLAB). Using the mean activity averaged during the same time window for trial periods as in the selectivity index (SI) (baseline, sample, early delay, late delay, and feedback periods), we computed the proportion of each type of trial classified correctly. For the cross-modality decoding in Fig. 4g, we used tactile trials for training and auditory trials for testing (tactile-to-auditory decoding), and vice versa (auditory-to-tactile decoding)[66]. The decoding accuracy was averaged between the tactile-to-auditory and auditory-to-tactile decoding. For modality-action-plan decoding in the unimodal d2-AFC task (Fig. 4i), we trained a linear SVM decoder to classify 4 different trial types (left/right trials with tactile/auditory modalities) and tested the classification performance with 5-fold cross-validation. For all the decoding analyses, only correct trials were used.

## Generalized linear model (GLM)

We used a generalized linear model (GLM) to parametrically describe task-related activity of each neuron[42,67]. The z-scored activity of each neuron was binned at 133 ms (2 frames) ($zF(t)$) and modeled as the linear combination of various task events and licking responses:

$$zF(t) \sim \sum_{n=1}^{N_{variables}} X_n(t)\beta_n, \qquad (4)$$

where $X_n(t)$ and $\beta_n$ are the design matrix and model coefficient matrix for the *n*-th variable, respectively, and $N_{variables}$ denotes the total number of variables ( = 32 variables) including 5 task events (sample, early delay, late delay, test, and feedback) × 2 action plans (left and right trials) × 3 outcome types (correct, commission-error, omission-error trials), and left- and right-lick variables. The design matrix for each task event was represented as boxcar functions to indicate the occurrence of the task event at each time bin, covering the entire duration of the task event but not overlapping each other. The design matrix for left- and right-lick variables covered from −9-time bins to 2-time bins from each lick (corresponding to −1200 ms to +267 ms from each lick).

GLMs were fit with elastic-net regularization to suppress the influence of predictors that are highly correlated and do not improve the model's accuracy ('lassoglm' function in MATLAB), in which the alpha value specifies the weight of lasso versus ridge optimization (alpha was set to 0.05, where alpha = 0 corresponds to ridge regularization, alpha = 1 to lasso regularization). The model coefficients $\beta_n$ were fit by maximizing the likelihood with 4-fold cross-validation. To assess the significance of each variable in the design matrix, we fitted a new GLM model obtained by removing the variable of interest (reduced model) from the full model. If for a certain neuron the deviance in the reduced model was significantly larger than that in the full model ($p < 0.05$, likelihood ratio test, Bonferroni correction for multiple comparisons), that neuron was deemed significantly modulated by the removed variable. In particular, to be consistent with the original property of action-plan-selective delay neurons, we tested the reduced model by removing the early-delay or the late-delay variable and thereby counted the number of those neurons that exhibited significant modulation by 1) either left or right early-delay variable, or 2) either left or right late-delay variable ('Early delay', or 'Late delay', respectively, in Supplementary Fig. 4c).

## Similarity index in state space

To evaluate the similarity of the population activity patterns between the tactile and auditory trials, we first performed dimensionality reduction of the neural population imaged in each age group using a principal component analysis. To match the number of neurons across all age groups, we first randomly selected 200 neurons for each age group and generated trial-averaged activity traces using 200 ms-binned windows for four trial type combinations (left/right trials × tactile/auditory modalities, using only correct trials) in each neuron. The first 15 principal components (PCs) represented over 85% of the variance in all age groups and thus the following similarity measurements between trajectories were performed in the 15D space. The similarity measurements between the two modalities (tactile and auditory) were computed in each of the left and right trials as follows[66]:

$$Similarity(t) = \frac{Ft'(t) \cdot Fa'(t)}{norm(Ft'(t)) \cdot norm(Fa'(t))} \qquad (5)$$

where $Ft(t)$ and $Fa(t)$ represent the 15D state-space trajectories in the tactile and auditory trials respectively, $Ft'(t)$ and $Fa'(t)$ denote their time derivative, and *norm* represents the norm of the 15D state-space vector. The similarity(t) was averaged over the early and the late delay period (0–1000 ms, and 1000–2000 ms after delay onset, respectively) and then averaged between left and right trials to obtain the similarity index. The above procedures including the random selection of neurons were repeated 1000 times to obtain a distribution of the similarity index for Fig. 4f.

## Resting-state functional connectivity analysis

Preceding each task session associated with calcium imaging, a resting state imaging was performed for 10 min, in which the mice were sitting in the same behavioral apparatus without engaging in a task. To examine how the activity of each pair of neurons was correlated during the resting state, calcium events that reflect action potential were identified for each neuron. We first extracted a raw fluorescence trace ($F_{raw}$) for each neuron as the mean value of all pixels inside the corresponding ROI as determined by the imaging data in the subsequent task session. To correct for potential contamination from out-of-focus neuropil fluorescence, we subtracted from the raw fluorescence ($F_{raw}$) the average fluorescence over a 20-μm ring surrounding that ROI ($F_{np}$):

$$F_{subt}(t) = F_{raw}(t) - 0.67 \times F_{np}(t), \qquad (6)$$

where 0.67 is a correction factor. Using the image data containing blood vessels, we estimated the correction factor as the ratio between the average fluorescence over a blood vessel and the neuropil adjacent to that blood vessel, both subtracted by a DC offset given by the intensity of off-lens pixels[40,42]. After applying the baseline correction due to bleaching, the fluorescent traces were z-scored using whole traces in each ROI as described above in "Imaging data analysis".

Using the extracted fluorescent trace for each neuron, candidate calcium events were detected whenever the amplitude crossed a threshold of 4 times median absolute deviations (MAD)[65]. We considered the events with an indicator decay time for GCaMP6f equal to or longer than 200 ms as real calcium events[40]. We chose the peaks of the transients to binarize calcium events[65] and then counted them within 100 ms bins. We finally computed the pairwise correlations using 10-min calcium event traces as the resting state functional connectivity (RSFC). To examine whether the RSFC depends on the functional properties of neurons, we computed the RSFC for two different pairs: 1) 'Task-relevant' pairs, where we computed the RSFC between pairs of action-plan-selective delay neurons that had the same sign of significant SI (selectivity index) either during the early or late delay period (termed 'task-relevant' RSFC). 2) 'Non-task-relevant' pairs, where we collected non-task-relevant neurons that had no significant SI during any task periods and computed the RSFC between pairs of non-task-relevant neurons and action-plan-selective delay neurons (termed 'non-task-relevant' RSFC). In Fig. 5b, the RSFC was plotted against the inter-neuronal distance (distance between the ROI centroids), binned at 50 μm. We only included neuron pairs with distances up to 400 μm, since only a small minority of fields of view had pairs separated by longer distances.

To estimate the significance of the observed RSFCs, we shuffled the calcium events of each neuron among the time bins thereby disrupting the temporal structure of spontaneous activity while preserving the overall activity level of each neuron. Using the shuffled data, the task-relevant and non-task-relevant RSFCs were similarly computed as stated above, which was repeated 1000 times to obtain a distribution of shuffled RSFCs for Fig. 5b, c.

## Functional connectivity analysis during task performance

Functional connectivity (FC) during task performance was calculated using binarized calcium events with 100 ms bins, similar to the RSFC analysis as described above. For the delay period (2 s) and during the baseline period (2-s time window preceding the sample period), we first generated the peri-stimulus time histogram (PSTH) for each trial type in each neuron. The PSTH was then subtracted from each calcium event trace to generate the residual activity trace. The pairwise correlations of the residual activity between neurons were computed as the FC. For Supplementary Fig. 11, the FC between pairs of action-plan-selective delay neurons that had the same sign of significant SI either during the early or late delay period was computed for the delay period and the baseline period in the bimodal task, and plotted against

the inter-neuronal distance binned at 50 μm. For Supplementary Fig. 12, FC in the unimodal task was calculated for 2 categories of neurons: 1) 'bimodal' neurons that had a significant and the same sign of SI during the delay period in the tactile and auditory trials, and 2) 'unimodal' neurons that had a significant SI only in either the tactile or auditory trials and did not belong to bimodal neurons (named as 'tactile-unimodal' neurons or 'auditory-unimodal' neurons, respectively). For the FC between unimodal neurons, we computed the FC between pairs of tactile-unimodal neurons and pairs of auditory-unimodal neurons.

## Spike sorting and electrophysiology data analysis

Neural data were filtered at 300–3000 Hz and re-referenced by a LCAR (local common average reference, inner radius: 60 μm, outer radius: 120 μm) implementation in SpikeInterface Python package[68]. Spikes were then sorted using HerdingSpikes (https://github.com/mhhennig/HS2)[69] and then manually curated using Phy (https://github.com/cortex-lab/phy). Manual curation was to ensure that the clusters do not contain noise and are well isolated based on visual guides provided by Phy graphical user interface (such as amplitude and features view over time, feature scatterplots inter-spike interval (ISI) distributions and correlograms, firing rates, waveforms, and waveform localization). Occasionally clusters needed to be manually separated from each other whenever informative features were visible and amplitude drifts patterns and ISI distribution suggested a separable multi-unit activity. Besides, two or more clusters were merged if the spike localization, features, and amplitude drift patterns suggested that they belong to the same neuron. After manual curation, a set of quality metrics were calculated using qualitymetrics module in SpikeInterface to evaluate the results of the spike sorting. Spike clusters were considered single units if they met criteria with ISI violation rate (number of ISIs smaller than 2 ms divided by the total number of ISIs) < 0.05 and signal-to-noise ratio > 6.0. To differentiate broad-spiking neurons (BS) from narrow-spiking (NS) neurons, we performed clustering analysis based on a Gaussian Mixture Model (GMM) fitted to the first 2 principal components of the mean spike waveforms during the repolarizing phase of spikes which is the best predictor of the cell-types based on optogenetic tagging studies[70–72]. To be consistent with calcium imaging targeting excitatory neurons, we used only BS neurons as putative pyramidal neurons for the following population analyses (949 BS neurons, 88.1% out of 1077 neurons, $n = 6$ mice, 17 sessions; Supplementary Fig. 6b).

As described for calcium imaging data ("Action plan selectivity"), we computed the selectivity index (SI) for action plan in each BS neuron based on spiking activity for each of the early delay and late delay (0–750 ms and 750–1500 ms from the delay onset, respectively; 1500 ms delay was used for the electrophysiology experiment). Using only correct trials, the trial-by-trial activity in left and right trials was collected to construct a ROC curve. Using the area under the ROC curve (AUC), the SI was computed as $2 \times (AUC - 0.5)$. We also performed single-trial decoding for left vs. right trials on a pseudo-population assembled across sessions as stated in "Decoding analysis" above. To estimate the variance of the decoding performance, on each iteration of the decoder (100 iterations), we randomly selected 200 neurons in each age group. We matched the number of trials per trial type (20 trials each for left and right trials) contributed by each neuron that was selected to participate in the population decoding. Using z-scored spiking activity during the early and late delay period for each neuron, we performed 5-fold cross-validation with a linear SVM decoder to estimate the classification performance and computed the proportion of each type of trial classified correctly.

## Optogenetic data analysis

The statistical significance of the difference in behavioral performance between laser-on and laser-off conditions was examined using a

bootstrapped permutation[24] in Fig. 6b, c, e, f, and Supplementary Fig. 14b, c. To test the laser effect in each age group, we randomly relabeled each trial regarding 1) laser-on or laser-off, 2) animals; 3) sessions; in each age group and computed the mean effect size averaged across mice (correct rate in the laser-off condition minus laser-on condition) in each permutation. For the comparison of the effect size between the age groups, we randomly relabeled each trial regarding 1) laser-on or laser-off, 2) animals; 3) sessions; across both age groups and computed the difference in the mean effect size between the age groups in each permutation. Each type of permutation was repeated 10,000 times to obtain the null distribution of the laser-induced behavioral changes. The *p*-value of the observed difference was calculated as a percentile of the null distribution.

### Statistics

Statistical analyses are described in the main text and the figure legends. Statistical analyses were performed using MATLAB. All statistical tests were two-sided unless otherwise stated. No statistical method was used to pre-determine sample sizes but our sample sizes are similar to those reported in previous publications[21,42].

### Reporting summary

Further information on research design is available in the Nature Portfolio Reporting Summary linked to this article.

## Data availability

The data generated in this study have been deposited on Zenodo (https://zenodo.org/record/8435602). Source data are provided with this paper.

## Code availability

Behavior data were acquired with Presentation (23.0) (Neurobehavioral Systems). Calcium imaging data were acquired using Inscopix Data Acquisition Software (IDAS v1.3.0, 1.7.1). Spike data were acquired using Intan RHX Data Acquisition Software (version 3.0.4). Behavior, imaging, and spike data were analyzed using MATLAB (R2019b, R2022a). Code used to process and analyze data is available on Zenodo (https://zenodo.org/record/8435602). Imaging data were pre-processed using Inscopix Data Processing Software (IDPS v1.8.0). Spike sorting was performed using HerdingSpikes (https://github.com/mhhennig/HS2) and Phy (https://github.com/cortex-lab/phy).

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

## Acknowledgements

This research is supported by the Ministry of Education, Singapore, under its MOE AcRF Tier 2 Award MOE-T2EP30220-0003, MOE AcRF Tier 1 Award RS01/19, MOE AcRF Tier 1 Award RG19/21, and by the Lee Kong Chian School of Medicine - Ministry of Education Start-Up Grant for T.K. We thank G. Augustine for providing CaMKIIα-Cre mouse line, S.Q. Tsong for animal husbandry, and V. Jayaraman, R.A. Kerr, D.S., Kim, L.L. Looger, and K. Svoboda from the GENIE Project for providing GCaMP6f.

## Author contributions

H.R.C., Y.R., M.Z.H.H., X.O., and T.K. were involved in conducting all the experiments and writing the manuscript. Y.R and T.K. analyzed the data. T.K. conceived, designed, and supervised the experiments.

## Competing interests
The authors declare no competing interests.
