## [Peer Review File · Nature Communications]

Reviewers' Comments:

Reviewer #1:

Remarks to the Author:

The study examined the neural mechanism underlying decline of executive function in aging. Using calcium imaging and optogenetic manipulation during memory-guided behavior, the authors show that working-memory coding and functional connectivity in the mouse medial prefrontal cortex (mPFC) are altered in aging. The authors examined the population activity of mPFC neurons in tactile and auditory cross-modal WM tasks and found exciting differences in coding ability of mPFC neurons of different age groups. Interestingly, switching from bimodal to single-modal task design revealed differences in model-general coding ability of mPFC neurons. Resting-state functional connectivity, especially among memory-coding neurons, decreased already in middle age. Optogenetic inactivation revealed that the middle-aged mPFC exhibited heightened vulnerability to perturbations. I think that the behavioral design of having multi- and uni-modality (auditory and somatosensory) involved is nice. The data collection processes are challenging, especially for the advanced age group. The results of this study could therefore be highly significant to the fields of aging and cognition. Several potentially confounding factors should be cautiously addressed by more experiments and analyses before the publication, though.

Major issues:

1. Are left vs. right licking rates already different during delay period or only begin to differ during the response period? Is there a difference among mice of different age groups in the premature licking during delay period? These are very important messages that readers need to know to assess the impact of the results. The GLM analysis in calcium signals was not sufficient to persuade away the worry that the licking during the delay period could already predict the output during response period, thus the motor act, besides motor memory, could influence the behavioral performance. If there is indeed biased licking during the delay period, more cautions should be added in the discussion.
2. Reward size is powerful in determine behavioral performance. Besides cognitive ability, a decline of reward processing could also influence the changes in performance through aging. Is the relative or subjective reward value the same for the mice of different age groups? For example, one could argue that middle and age mice weight more and need more daily water, thus the relative or subjective reward size is smaller for them. Such difference in relative or objective value may impact on behavior. The authors could perform experiments with varying rewards to clarify it.
3. Are the expression levels of virus/GCaMP similar for the mice of different age groups? Is the density of infected cells similar for these groups? Are the changes in $\Delta F/F$ represent similar firing rate change? The authors should perform further experiments and analyses to address these questions, which potentially could impact the validity of the conclusions. For the last question, electrophysiological recording in vivo or in brain slices could be made.
4. Is functional coupling during the baseline period important for behavioral performance or simply a byproduct? Are there differences in coupling for bi-modal and uni-modal neurons? Are the results of correlation analysis during the delay period also showed the similar results? More analyses and discussion should be added to address these questions.

Minor issues:

1. Detailed information of age groups is important for the study and should be included in the results part. Now readers must go to method section for it.
2. When comparing decoding results, the authors should use the same number of neurons. Typically, it could be done by use the minimal number of neurons for different groups.
3. In the title, 'cognitive aging' seems a strange expression. The study is more about 'cognitive decline in aging'.

Reviewer #2:

Remarks to the Author:

Chong et al. describe how the excitatory neurons in the mPFC deteriorate in storing modality-general and modality-specific working memory as mice age. They assessed the memory-based performance of mice in the d2-AFC and imaged mPFC neuronal activity during the task in three

age groups: young, middle-aged, and advanced-aged groups. The behavioral results are expected, as aging is a recognized component that degrades memory-guided behaviors in both humans and rodents. They did, however, investigate the neural underpinnings of such deterioration by examining modality-general or modality-specific selectivity in the delayed activity (action-plan coding) of mPFC neurons during the task.

Their data appear to be straightforward, and the manuscript is well-written. Measuring resting-state functional connectivity (RSFC) was especially intriguing because it obviously supported the notion that neural activity linked to the memory-based action plan weakens as the animal ages.

However, I have a major concern about their claim that the modality-general coding declines with age. They only trained 5 advanced-aged mice (18-23 months old). Most mice at this age cannot hear well (B6 mice start to show hair-cell loss when they become 6 months old). It is clear that the mice cannot perform the auditory task without hearing. Indeed, it is not determined by the authors whether aging hampered modality-general processing in the mPFC or simply caused hearing loss. It is very likely that the aged mice use only tactile cues from the learning stage (which may explain why the old mice learned slowly), and the mice cannot perform the task only with auditory unisensory cues (which was shown in Fig. 3). If the authors want to argue cross-modal memory coding in the mPFC, they must use sensory modalities that can be well sensed at the same level as tactile left/right stimuli.

Furthermore, there are two different sub-groups of mice in the middle-aged group: one with greater correct rates (above 80%) and the other with lower correct rates (below 80%). Because there are two different groups that exhibit greater or lower correct rates in auditory blocks, it is conceivable that the latter mice might lose their hearing ability more than the former.

Mice cannot create auditory memories in the absence of auditory inputs to the brain (hearing loss due to hair-cell loss). If the authors want to address crossmodal memory, they must use other sensory modalities that can be well detected at a comparable level to tactile stimuli they used across the tested ages in mice. I strongly recommend using visual cues instead of auditory cues (I believe the old mice can still detect flashlights in darkness), spatially separated like the tactile cues they used, to determine whether the modality generalization for the action plan after learning in the mPFC, rather than just simple sensory detection, degrades with aging.

Reviewer #1 (reviewer's comments in italic):

The study examined the neural mechanism underlying decline of executive function in aging. Using calcium imaging and optogenetic manipulation during memory-guided behavior, the authors show that working-memory coding and functional connectivity in the mouse medial prefrontal cortex (mPFC) are altered in aging. The authors examined the population activity of mPFC neurons in tactile and auditory cross-modal WM tasks and found exciting differences in coding ability of mPFC neurons of different age groups. Interestingly, switching from bimodal to single-modal task design revealed differences in model-general coding ability of mPFC neurons. Resting-state functional connectivity, especially among memory-coding neurons, decreased already in middle age. Optogenetic inactivation revealed that the middle-aged mPFC exhibited heightened vulnerability to perturbations. I think that the behavioral design of having multi- and uni-modality (auditory and somatosensory) involved is nice. The data collection processes are challenging, especially for the advanced age group. The results of this study could therefore be highly significant to the fields of aging and cognition. Several potentially confounding factors should be cautiously addressed by more experiments and analyses before the publication, though.

We thank the reviewer for all the thoughtful comments and evaluations. We provide our responses to the reviewer's comments below.

Major issues:

1. Are left vs. right licking rates already different during delay period or only begin to differ during the response period? Is there a difference among mice of different age groups in the premature licking during delay period? These are very important messages that readers need to know to assess the impact of the results. The GLM analysis in calcium signals was not sufficient to persuade away the worry that the licking during the delay period could already predict the output during response period, thus the motor act, besides motor memory, could influence the behavioral performance. If there is indeed biased licking during the delay period, more cautions should be added in the discussion.

In response to the reviewer's question, we examined whether left vs. right lick rates were already different during the delay period between left and right trials (Reviewer Fig. 1, included as Supplementary Fig. 4d). We found that premature licks occurred in about half of the trials and were predominantly directed to the correct side and that there was a significant bias to left licking on left trials, and right licking on right trials ($P < 0.001$ for both left and right trials; the main effect of licking direction, two-way ANOVA (licking direction x age group)). However, there was no significant difference among the age groups ($P = 0.83$ and 0.65 for left and right trials respectively; the main effect of age group, two-way ANOVA (licking direction x age group)), nor any interactions between the age groups and the licking directions ($P = 0.89$ and 0.62 for left and right trials respectively). Therefore, premature licking does not account for the differences in neuronal activity across age groups.

Reviewer Figure 1. Mean lick rates during the delay period on left and right trials. Mean lick rate during the delay period of left and right trials in the bimodal task averaged across all calcium imaging sessions (young, $n = 22$ sessions, 11 mice; middle age, $n = 14$ sessions, 7 mice; advanced age, $n = 10$ sessions, 5 mice). Error bars, \pm s.e.m. Y, young; M, middle age; A, advanced age.

In addition, to examine how such premature licking could affect our main results of delay activity, we collected only trials with no premature licking during the delay period (which corresponded to 49.3% and 65.4% of total correct trials for bimodal and unimodal tasks, respectively) and applied the same population analyses as in the main Figs. 2 and 4. Decoding accuracy during the delay declined as a function of age in the bimodal task (Reviewer Fig. 2, included as Supplementary. Fig. 4e), which is consistent with the original results shown in Fig. 2g. We also confirmed that modality-general coding degraded with aging in the uni-modal task by evaluating the similarity index in state space and cross-modality decoding (Reviewer Fig. 3a–c, included as Supplementary. Fig. 9c–e), which replicated the original main results shown in Fig. 4f–h. These results further support that the lick behavior was similar across groups, and does not affect our main conclusions. We incorporated these results into Supplementary Figures 4 and 9 and the text (page 3, 2nd line from the bottom–page 4, 1st paragraph; page 6, 4th paragraph).

In accordance with the reviewer's comment, we acknowledge the significance of addressing the issue of premature licking during the delay period. As a response, we have included a dedicated discussion to clarify this matter (page 9, 2nd paragraph).

Reviewer Figure 2. Decoding accuracy in trials with no premature licks during the delay in the bimodal task. Decoding accuracy of action plan in each of the task periods using 300 randomly selected neurons for each age group. Decoding accuracy during the early and late delay periods (gray shading) in the middle-aged and the advanced-aged groups was significantly lower than that in the young group ($P < 0.001$, two-way ANOVA (period \times age group) with Bonferroni post hoc test). Error bars, \pm s.e.m. across 100 iterations. The horizontal dotted lines show the 97.5th and 2.5th percentiles of the shuffled data. Bsl, baseline; Smp, sample; eDly, early delay; IDly, late delay; Fbc, feedback period.

Reviewer Figure 3. Age-dependent modality-general delay activity in trials with no premature licks during the delay. **a**, Similarity index in state space. $**P < 0.001$, one-way ANOVA with Bonferroni post hoc test. Error bars, \pm s.d. across 1,000 iterations. **b**, Classification accuracy of action plan using within-modality and cross-modality decoding with 200 randomly selected neurons for each age group. Horizontal lines, 97.5th and 2.5th percentile of the shuffled distribution. Error bars, \pm s.e.m. across 100 iterations. **c**, Product of multiplying within-modality and cross-modality decoding accuracy. $**P < 0.001$, one-way ANOVA with Bonferroni post hoc test. Error bars, \pm s.e.m. across 100 iterations.

2. Reward size is powerful in determine behavioral performance. Besides cognitive ability, a decline of reward processing could also influence the changes in performance through aging. Is the relative or subjective reward value the same for the mice of different age groups? For example, one could argue that middle and age mice weight more and need more daily water, thus the relative or subjective reward size is smaller for them. Such difference in relative or objective value may impact on behavior. The authors could perform experiments with varying rewards to clarify it.

We appreciate the reviewer’s suggestions. To investigate the influence of relative reward value on behavioral performance across different age groups, we conducted experiments employing a tactile-based delayed 2-AFC task with two reward size conditions: default and high-reward conditions. In the high-reward condition, the reward size was twice that of the default condition (typically 1 μ l vs. 2 μ l of water per correct trial for default vs. high-reward). Despite the variations in reward size, we did not observe any systematic changes in behavioral performance across all age groups ($P = 0.41$, the main effect of reward size, two-way ANOVA of reward size x age group; Reviewer Fig. 4a, included as Supplementary Fig. 5a). Also, there was no significant effect of age group nor their interaction ($P = 0.44$ and 0.53 , for the main effect of age group and interaction, respectively). We suggest that the default reward size was sufficient to motivate the animals in all age groups to actively engage in the task, rendering any additional increase in reward size ineffective in further improving behavioral performance.

Furthermore, we examined whether the variations in reward size had an age-dependent effect on the action-plan selectivity of neurons (Reviewer Fig. 4b, included as Supplementary Fig. 5b). Our analysis revealed that the absolute selectivity index (|SI|) during the early and the late delay period did not exhibit any significant dependence on reward size across age groups ($P > 0.2$ and $P > 0.5$, the main effect of reward size and its interaction with age group, respectively;

two-way ANOVA). Nonetheless, there was a notable age-dependent decrease in $|SI|$, consistent with our original findings ($P < 0.001$, main effect of age group, two-way ANOVA). These findings suggest that the age-dependent alteration in the action-plan selectivity is unlikely to be solely attributed to variations in reward processing. We added these results in Supplementary Figure 5 and in the text (page 4, 2nd paragraph).

Reviewer Figure 4. Impact of reward size on behavior and action-plan selectivity. **a**, Correct rate in the tactile d2-AFC task with a variable reward size (young, $n = 10$ default and 8 high reward sessions, 5 mice; middle age, $n = 10$ default and 7 high reward sessions, 5 mice; advanced age, $n = 7$ default and 6 high reward sessions, 4 mice). Open circles, individual sessions. Def, default reward condition, High, high reward condition. Box plots show the median and the 25th and 75th percentiles as box edges and the whiskers extend to the 5th and 95th percentiles. Y, young; M, middle age; A, advanced age. **b**, Mean absolute selectivity index for action plan ($|SI|$) (young, $n = 1,045$ and 659 neurons; middle age, $n = 407$ and 529 neurons; advanced age, $n = 407$ and 376 neurons; for Def and High reward conditions, respectively). Error bars, \pm s.e.m. * $P < 0.005$, ** $P < 0.001$, two-way ANOVA (reward size x age group) with Bonferroni post hoc test.

3. Are the expression levels of virus/GCaMP similar for the mice of different age groups? Is the density of infected cells similar for these groups? Are the changes in $\Delta F/F$ represent similar firing rate change? The authors should perform further experiments and analyses to address these questions, which potentially could impact the validity of the conclusions. For the last question, electrophysiological recording in vivo or in brain slices could be made.

To evaluate the expression levels of GCaMP, we measured the fluorescent intensity of GCaMP6f in histology sections of the imaged mice (Reviewer Fig. 5a, Supplementary Fig. 2d). There was no significant difference among the age groups ($P = 0.95$, one-way ANOVA). In addition, we also measured the amplitude of $\Delta F/F$ of calcium events during the resting state, but found no age-dependent difference (Reviewer Fig. 5b, Supplementary Fig. 2e; $P = 0.76$, one-way ANOVA). Next, to quantify the density of infected cells in each age group, we counted

GCaMP6f-positive cells in histology sections of the mice (Reviewer Fig. 5c, Supplementary Fig. 2f). There was no significant difference among the age groups ($P = 0.91$, one-way ANOVA). These results collectively indicate that differences in the expression levels of GCaMP6f, the amplitude of the fluorescent signal, or the density of infected neurons are unlikely to account for the age-dependent deteriorations observed in the task-relevant activity. We added these results in Supplementary Fig. 2d–f and in the text (page 4, 3rd paragraph).

Addressing the reviewer's question regarding whether the GCaMP signal changes in calcium imaging represent similar firing rate changes, we examined whether a consistent age-dependent decline in action-plan selectivity could be observed in spiking activity. We conducted electrophysiological recording while mice were performing the tactile-based delayed 2-AFC (d2-AFC) task. The behavioral performance was comparable across age groups (Reviewer Fig. 6a, Supplementary Fig. 6a). We identified two distinct populations, broad-spiking neurons (BS) and narrow-spiking (NS) neurons (Reviewer Fig. 6b, Supplementary Fig. 6b). To be consistent with calcium imaging targeting excitatory neurons, we analyzed only BS neurons as putative pyramidal neurons. As observed in calcium imaging, we found consistent and similar age-dependent decreases in the absolute selectivity index for action plan (left/right) (|SI|) and the decoding accuracy of action plan for both the early and the late delay period ($P < 0.001$ for all, one-way ANOVA with age group) (Reviewer Fig. 6c,d, Supplementary Fig. 6c,d). These findings provide further support and validation for the calcium imaging results, suggesting that the observed signal changes in calcium imaging reflect consistent changes in firing rates across different age groups. We described these results in the text (page 4, last paragraph–page 5, 1st paragraph) and Supplementary Fig. 6.

Reviewer Figure 5. Expression level and intensity of GCaMP6f. **a**, Fluorescent intensity of GCaMP6f in histology sections ($n = 11$ young mice, $n = 7$ middle-aged mice, $n = 5$ advanced-aged mice). The mean fluorescent intensity of GCaMP6f was calculated as the mean pixel intensity of GCaMP6f-positive cells normalized by the mean background intensity outside the brain slice. **b**, Amplitude of $\Delta F/F$ of calcium events during the resting state. The maximum dF/F amplitude across all calcium events was calculated for each neuron, and then the median was computed across all the imaged neurons. **c**, Density of GCaMP6f-positive cells in histology sections of the imaged mice. Box plots show the median and the 25th and 75th percentiles as box edges and the whiskers extend to the 5th and 95th percentiles. Open circles, individual mice. Y, young; M, middle age; A, advanced age.

Reviewer Figure 6. Age-dependent decline in action-plan selectivity based on spiking activity. **a**, Correct rate during the electrophysiology recording sessions with the tactile d2-AFC task (young, $n = 5$ sessions, 2 mice; middle age, $n = 6$ sessions, 2 mice; advanced age, $n = 6$ sessions, 2 mice). Open circles, individual sessions. NS, no significant difference among age groups ($P > 0.3$ for all comparisons, two-sided Wilcoxon rank-sum test). Box plots show the median and the 25th and 75th percentiles as box edges and the whiskers extend to the 5th and 95th percentiles. Y, young; M, middle age; A, advanced age. **b**, Distribution of spike width (trough-to-peak interval) of all recorded neurons. Blue and gray denote Broad-spiking (BS) and narrow-spiking (NS) neurons, respectively. Inset, the mean spike waveform for BS (blue) and NS (gray) neurons. The spike amplitude was normalized by its trough-peak amplitude for each neuron. Shadings, \pm s.e.m. **c**, Mean absolute selectivity index for action plan ($|SI|$) (young, $n = 225$ neurons; middle age, $n = 352$ neurons; advanced age, $n = 362$ neurons). Error bars, \pm s.e.m. ****** $P < 0.001$, one-way ANOVA with Bonferroni post hoc test. **d**, Decoding accuracy of action plan in each of the delay periods using 200 randomly selected neurons for each age group (Methods). Decoding accuracy during the early and late delay periods in the middle-aged and the advanced-aged groups was significantly lower than that in the young group ($P < 0.001$, one-way ANOVA with Bonferroni post hoc test). Error bars, \pm s.e.m. across 100 iterations. Horizontal lines, 97.5th and 2.5th percentile of the shuffled distribution.

4. Is functional coupling during the baseline period important for behavioral performance or simply a byproduct? Are there differences in coupling for bi-modal and uni-modal neurons? Are the results of correlation analysis during the delay period also showed the similar results? More analyses and discussion should be added to address these questions.

We appreciate the reviewer's questions and suggestions. To answer them, we first analyzed the activity during the delay period of the bimodal task, as well as during the baseline period, that is 2 sec preceding the sample stimulus presentation, and computed the functional connectivity (FC) – i.e., pairwise correlations between action-plan-selective delay neurons – as did in our original Figure 5 during the resting state outside of the task context. For computing FC, we subtracted the corresponding peri-stimulus time histogram (PSTH) from each calcium event train, which was generated for each neuron and trial type. We confirmed that the FC during the task declined as a function of age ($P < 0.0001$, the main effect of age group, three-way ANOVA (age group x task period x interneuronal distance)) (Reviewer Fig. 7a, Supplementary Fig. 11a), consistent with the resting state functional connectivity (RSFC) in Figure 5b.

To further examine the FC's importance for behavioral performance, FC was compared between correct trials and error trials (Reviewer Fig. 7b,c, Supplementary Fig. 11b,c). The FC during the delay period between action-plan-selective delay neurons was significantly larger in correct trials than in error trials only in the young group ($P = 0.0048$, the main effect of behavioral performance, two-way ANOVA (behavioral performance x interneuronal distance)), but not in other age groups ($P > 0.3$). The FC during the baseline period was also significantly larger in correct trials than in error trials only in the young group ($P = 0.012$), but not in other age groups ($P > 0.15$). These results suggest that the FC between action-plan-selective delay neurons both during the baseline and delay periods was predictive of the upcoming performance of working memory in the young group, but such predictive functional coupling is no longer detectable in middle-aged and advanced-aged groups. Together with the resting state functional connectivity shown in Figure 5, our results revealed the age-dependent decline in the task-relevant network both during task engagement and outside of the task context.

Next, to answer the reviewer's question about the differences in coupling for bimodal and unimodal neurons, we examined the FC in the unimodal task. We defined 'bimodal' neurons as those that had a significant and the same sign of selectivity index (SI for left vs. right) during the delay period in the tactile and auditory trials, and 'unimodal' neurons as those that had a significant SI only in either the tactile or auditory trials and did not belong to bimodal neurons (named as 'tactile-unimodal' neurons or 'auditory-unimodal' neurons, respectively). For the FC between unimodal neurons, we computed the FC between the same modality neurons, that is, pairs of tactile-unimodal neurons and pairs of auditory-unimodal neurons. During the delay period, bimodal neurons showed significantly higher FC compared to unimodal neurons in the young group ($P = 0.010$, the main effect of neuron type, two-way ANOVA (neuron type x interneuronal distance)) and only weakly in the middle-age ($P = 0.042$), but not in the advanced age group ($P = 0.47$) (Reviewer Fig. 8a, Supplementary Fig. 12a). During the baseline period, bimodal neurons exhibited significantly higher FC than unimodal neurons in the young adults ($P = 0.016$), but not in the other age groups ($P > 0.6$) (Reviewer Fig. 8b, Supplementary Fig. 12b). These findings suggest that bimodal neurons exhibit stronger functional coupling with each other, forming a more robust functional network compared to unimodal neurons in young adults. However, the strength of the bimodal neuron network deteriorates with age. We added Supplementary Figs. 11 and 12, descriptions to describe these results (page 7, last paragraph – page 8, 1st paragraph), and discussion in the text (page 9, last paragraph – page 10, 1st paragraph).

Reviewer Figure 7. Functional connectivity (FC) during the bimodal tasks. **a**, FC during the delay (left) and baseline (right) periods in the bimodal task as a function of the inter-neuronal distance. **b**, FC in correct trials (blue) and error trials (cyan) was computed between pairs of action-plan-selective delay neurons that had the same sign of action-plan selectivity for each imaging session. Only sessions with 50 or more error trials were included (young, $n = 6,584$ pairs across 441 cells from 9 sessions; middle age, $n = 1,646$ pairs across 175 cells from 7 sessions; advanced age, $n = 3,925$ pairs across 343 cells from 8 sessions). **c**, FC during the baseline period of the same trials and pairs as in **b**. Error bars denote \pm s.e.m.

Reviewer Figure 8. Functional connectivity (FC) for bimodal and unimodal neurons. a, FC during the delay period in the unimodal task as a function of the inter-neuronal distance. FC for bimodal neurons (blue) was computed between pairs of action-plan-selective delay neurons that had a significant and the same sign of selectivity index (SI) during the delay period in the tactile and auditory trials (young, $n = 1,882$ pairs across 318 cells from 16 sessions; middle-age, $n = 266$ pairs across 87 cells from 13 sessions; advanced-age, $n = 75$ pairs across 42 cells from 9 sessions). FC for unimodal neurons (green) was computed between pairs of action-plan-selective delay neurons that had a significant SI either in the tactile or auditory trials and did not belong to bimodal neurons (young, $n = 4,597$ pairs across 694 cells; middle-age, $n = 2,752$ pairs across 456 cells; advanced-age, $n = 1,984$ pairs across 348 cells). **b,** FC during the baseline period of the same trials and pairs as in a. Error bars denote \pm s.e.m.

Minor issues:

1. Detailed information of age groups is important for the study and should be included in the results part. Now readers must go to method section for it.

We thank the reviewer for pointing this out. We have described the detailed information of age groups in the Results (page 2, last paragraph).

2. When comparing decoding results, the authors should use the same number of neurons. Typically, it could be done by use the minimal number of neurons for different groups.

We apologize for the confusion caused by our unclear description of the results. We used the same number of neurons when comparing the decoding performance among groups in Fig. 2g (300 neurons for the bimodal task for each age group) and Fig. 4g–i (200 neurons for the unimodal task for each age group) as described in Methods, Decoding analysis. We now described this information in the results (page 3, 3rd paragraph; page 5, last paragraph) and the figure legends (Fig. 2g legend; Fig. 4g legend).

3. In the title, ‘cognitive aging’ seems a strange expression. The study is more about ‘cognitive decline in aging’.

We thank the reviewer for pointing this out. We believe that ‘cognitive aging’ is nowadays used to refer to the natural and gradual decline in cognitive function that occurs as people age, as can be seen well in some literature and websites (e.g., Andrews-Hanna et al., *Neuron* **56**, 924-935, 2007; Cabeza, R. et al. *Nature reviews. Neuroscience* **19**, 701-710, 2018; Hedden, T. & Gabrieli, J. D. *Nature reviews. Neuroscience* **5**, 87-96, 2004; “Cognitive ageing articles from across Nature Portfolio” <https://www.nature.com/subjects/cognitive-ageing>).

Nevertheless, we agree that the wording may be unfamiliar to readers and we thus added an explanation of this wording in the Introduction (page 2, 1st paragraph).

References

Andrews-Hanna, J. R. et al. Disruption of large-scale brain systems in advanced aging. *Neuron* **56**, 924-935, doi:10.1016/j.neuron.2007.10.038 (2007).

Cabeza, R. et al. Maintenance, reserve and compensation: the cognitive neuroscience of healthy ageing. *Nature reviews. Neuroscience* **19**, 701-710, doi:10.1038/s41583-018-0068-2 (2018).

Hedden, T. & Gabrieli, J. D. Insights into the ageing mind: a view from cognitive neuroscience. *Nature reviews. Neuroscience* **5**, 87-96, doi:10.1038/nrn1323 (2004).

Reviewer #2 (reviewer's comments in italic):

Chong et al. describe how the excitatory neurons in the mPFC deteriorate in storing modality-general and modality-specific working memory as mice age. They assessed the memory-based performance of mice in the d2-AFC and imaged mPFC neuronal activity during the task in three age groups: young, middle-aged, and advanced-aged groups. The behavioral results are expected, as aging is a recognized component that degrades memory-guided behaviors in both humans and rodents. They did, however, investigate the neural underpinnings of such deterioration by examining modality-general or modality-specific selectivity in the delayed activity (action-plan coding) of mPFC neurons during the task.

Their data appear to be straightforward, and the manuscript is well-written. Measuring resting-state functional connectivity (RSFC) was especially intriguing because it obviously supported the notion that neural activity linked to the memory-based action plan weakens as the animal ages.

However, I have a major concern about their claim that the modality-general coding declines with age. They only trained 5 advanced-aged mice (18-23 months old). Most mice at this age cannot hear well (B6 mice start to show hair-cell loss when they become 6 months old). It is clear that the mice cannot perform the auditory task without hearing. Indeed, it is not determined by the authors whether aging hampered modality-general processing in the mPFC or simply caused hearing loss. It is very likely that the aged mice use only tactile cues from the learning stage (which may explain why the old mice learned slowly), and the mice cannot perform the task only with auditory unisensory cues (which was shown in Fig. 3). If the authors want to argue cross-modal memory coding in the mPFC, they must use sensory modalities that can be well sensed at the same level as tactile left/right stimuli.

Furthermore, there are two different sub-groups of mice in the middle-aged group: one with greater correct rates (above 80%) and the other with lower correct rates (below 80%). Because there are two different groups that exhibit greater or lower correct rates in auditory blocks, it is conceivable that the latter mice might lose their hearing ability more than the former.

Mice cannot create auditory memories in the absence of auditory inputs to the brain (hearing loss due to hair-cell loss). If the authors want to address crossmodal memory, they must use other sensory modalities that can be well detected at a comparable level to tactile stimuli they used across the tested ages in mice. I strongly recommend using visual cues instead of auditory cues (I believe the old mice can still detect flashlights in darkness), spatially separated like the tactile cues they used, to determine whether the modality generalization for the action plan after learning in the mPFC, rather than just simple sensory detection, degrades with aging.

We appreciate the reviewer for pointing out this issue. In accordance with the reviewer's suggestion, we conducted training for a visuo-tactile version of the d2-AFC task. This task involved a bimodal stimulus, combining a visual stimulus (a white LED turned on in either the left or right visual field) with a tactile stimulus (air puffs to either the left or right cheek). During this task, when an air puff was directed to the left cheek along with the left LED, it was referred to as a 'left' trial, and similarly for the right. After training mice with the visuo-tactile bimodal d2-AFC task to achieve 70% correct rate, the mice were further trained and tested with the visuo-tactile version of the unimodal d2-AFC task, in which we presented visual, tactile, or bimodal

stimulus in interleaved blocks of 20–30 trials each, akin to the original audio-tactile d2-AFC task. Despite undergoing extensive unimodal training for over a month ($n = 38.3 \pm 7.2$ training sessions, mean \pm s.e.m. across mice) in addition to the bimodal task training, the mice encountered challenges in learning the visual d2-AFC task with a 2-s delay. In contrast, their performance remained consistently robust in both the bimodal and tactile tasks (Reviewer Fig. 9). Due to the consistently subpar performance of the visual task compared to the tactile task across all age groups, the feasibility of examining neural activity related to both visual and tactile modalities became impractical. Consequently, this particular aspect was not included in the manuscript.

Reviewer Figure 9. Mean correct rate in the visuo-tactile d2-AFC task averaged across unimodal training sessions ($n = 5$ young mice, $n = 3$ middle-aged mice, $n = 5$ advanced-aged mice). Box plots show the median and the 25th and 75th percentiles as box edges and the whiskers extend to the 5th and 95th percentiles. Y, young; M, middle age; A, advanced age.

Instead, we opted to analyze the data using the audio-tactile d2-AFC task. We concur with the reviewer's comment that advanced-aged mice exhibited compromised performance in auditory trials, indicating the presence of age-related hearing loss. It is therefore plausible that this hearing loss might disrupt the establishment of auditory-based representations, thus contributing to the observed reduction in crossmodal representation within the advanced-age group. To address a parallel concern within the middle-aged group, we collected additional imaging sessions, focusing exclusively on sessions where the middle-aged mice exhibited performance comparable to young mice (Reviewer Fig. 10a, Supplementary Fig. 8a). Even in these selective sessions, the middle-aged mice exhibited a degradation in modality-general coding when compared to the young mice (Reviewer Fig. 10b–e, Supplementary Fig. 8b–e). This result indicates that the decline in modality-general coding, at least within the middle-aged group, cannot be solely ascribed to hearing loss. In response to the reviewer's comments, we added these results in Supplementary Figure 8 and the text (page 6, 2nd paragraph) and engaged in a discussion about the potential impact of hearing loss (page 9, 3rd paragraph).

Reviewer Figure 10. Comparisons of modality-general activity using performance-matched middle-aged mice vs. young mice in the auditory task. **a**, Correct rate in the unimodal d2-AFC task as well as the bimodal task during the imaging sessions ($n = 10$ young mice, $n = 7$ middle-aged mice). NS, $P > 0.5$, two-sided Wilcoxon rank-sum test. Box plots show the median and the 25th and 75th percentiles as box edges and the whiskers extend to the 5th and 95th percentiles. Y, young; M, middle age; A, advanced age. Error bars, \pm s.e.m. $**P < 0.001$, one-way ANOVA. **b**, Selectivity indices (SIs) in the tactile and the auditory trials were multiplied for each neuron and averaged across all neurons (young, $n = 1,577$ neurons; middle age, $n = 813$ neurons). **c**, Similarity index in state space. $*P < 0.005$, $**P < 0.001$, one-way ANOVA. Error bars, \pm s.d. across 1,000 iterations. **d**, Classification accuracy of action plan using within-modality and cross-modality decoding with 200 randomly selected neurons for each age group. Horizontal lines, 97.5th and 2.5th percentile of the shuffled distribution. Error bars, \pm s.e.m. across 100 iterations. **e**, Product of multiplying within-modality and cross-modality decoding accuracy. $**P < 0.001$, one-way ANOVA with Bonferroni post hoc test. Error bars, \pm s.e.m. across 100 iterations.

Reviewers' Comments:

Reviewer #1:

Remarks to the Author:

My previous comments and concerns have been adequately addressed, therefore recommending to accept the manuscript. In summary, I think this exciting study provides important insights to the field of aging and working memory.

Reviewer #2:

Remarks to the Author:

I appreciate the authors' efforts to resolve my concern by doing further experiments and data analysis. It was surprising that even young mice were unable to learn the visual task. Instead of the visual test, they did further imaging in the middle-aged mice that demonstrated comparable auditory performance in the d2-AFC task. The authors argued that the modality-general working memory of middle-aged mice has deteriorated, despite their performance being comparable to that of young mice.

This new data suggests that the degradation of modality-general coding occurs earlier in the delay period than cognitive decline in working memory tasks. What would be then the meaning of changes in the mPFC neural activity throughout aging? If the authors assert that the mPFC activity changes happen before any behavioral indications of cognitive aging, this must be addressed in the text. The neural activity they measure may not cause but may correlate with cognitive loss, such as working memory impairment, which occurs after the neural activity changes.

I also want to point out that if the authors wanted to assess perceptual abilities rather than working memory, they should examine mice's performance in the 2-AFC task without delay. Middle-aged mice must perform well in the 2-AFC task but not in the d2-AFC task if their perceptual ability did not decline but the working memory did. Please discuss this issue thoroughly in the text.

As a minor point, it appears that the decline in decoding accuracy between within-modality and cross-modality was not different between young and middle-aged mice in the new supplementary Figure 8d. Is it still valid to stress that the modality-general coding of the working memory is degraded first in the mPFC? Please explain this.

Reviewer #1 (reviewer's comments in italic):

My previous comments and concerns have been adequately addressed, therefore recommending to accept the manuscript. In summary, I think this exciting study provides important insights to the field of aging and working memory.

We thank the reviewer for the positive evaluations.

Reviewer #2 (reviewer's comments in italic):

I appreciate the authors' efforts to resolve my concern by doing further experiments and data analysis. It was surprising that even young mice were unable to learn the visual task. Instead of the visual test, they did further imaging in the middle-aged mice that demonstrated comparable auditory performance in the d2-AFC task. The authors argued that the modality-general working memory of middle-aged mice has deteriorated, despite their performance being comparable to that of young mice.

This new data suggests that the degradation of modality-general coding occurs earlier in the delay period than cognitive decline in working memory tasks. What would be then the meaning of changes in the mPFC neural activity throughout aging? If the authors assert that the mPFC activity changes happen before any behavioral indications of cognitive aging, this must be addressed in the text. The neural activity they measure may not cause but may correlate with cognitive loss, such as working memory impairment, which occurs after the neural activity changes.

We thank the reviewer for highlighting this point. The comparable working-memory performance between the young and the middle-aged groups, at first glance, appears to implicate that the relevant neural circuit can still be properly working in middle age. However, we propose that beneath apparently normal behavioral performance in middle age, the underlying neural computations are trending towards suboptimal functioning and reduced resilience to perturbation, manifesting in observed alterations in mPFC neural activity and functional connectivity. Indeed, as depicted in Fig. 6d-f through optogenetic manipulation, the mPFC circuit of the middle-aged group exhibits heightened vulnerability to perturbation, despite the seemingly normal behavioral performance in the no-perturbation condition. Responding to the reviewer's comments, we have integrated a dedicated discussion to elaborate on this issue (page 9, 4th paragraph).

I also want to point out that if the authors wanted to assess perceptual abilities rather than working memory, they should examine mice's performance in the 2-AFC task without delay. Middle-aged mice must perform well in the 2-AFC task but not in the d2-AFC task if their perceptual ability did not decline but the working memory did. Please discuss this issue thoroughly in the text.

We understand the reviewer's point; incorporating the 2-AFC task without delay, alongside the d2-AFC task for middle-aged mice, will help discern whether any behavioral impairment stems from perceptual disabilities or working memory deficits. Nevertheless, we wish to underscore that the primary focus of our study lies in unveiling alterations in neural activity that manifest even in the absence of overt behavioral impairment during middle age, as elucidated in response to the reviewer's comment above. In light of this, we have incorporated this point in the discussion (page 9, 3rd paragraph).

As a minor point, it appears that the decline in decoding accuracy between within-modality and cross-modality was not different between young and middle-aged mice in the new supplementary Figure 8d. Is it still valid to stress that the modality-general coding of the working memory is degraded first in the mPFC? Please explain this.

We appreciate the reviewer's comments. In Supplementary Fig. 8d, we found that the difference in decoding accuracy between within-modality and cross-modality was significantly larger in the middle-aged group compared to the young, both in the early and late delay ($P = 0.011$ and 0.016 , respectively, Wilcoxon signed-rank test), which consistently indicates a deterioration in modality-general coding in middle age. We have added this in Supplementary Fig. 8d legend.